# Asymmetric REINFORCE for off-Policy Reinforcement Learning: Balancing positive and negative rewards

**Charles Arnal**
FAIR, MSL, Meta

**Gaëtan Narozniak**
FAIR, MSL, Meta

**Vivien Cabannes**
FAIR, MSL, Meta

**Yunhao Tang**
Anthropic (ex-Meta)

**Julia Kempe**
FAIR, MSL, Meta
NYU Courant Institute and CDS

**Rémi Munos**
FAIR, MSL, Meta

## Abstract

Reinforcement learning (RL) is increasingly used to align large language models (LLMs). Off-policy methods offer greater implementation simplicity and data efficiency than on-policy techniques, but often result in suboptimal performance. In this work, we study the intermediate range of algorithms between off-policy RL and supervised fine-tuning by analyzing a simple off-policy REINFORCE algorithm, where the advantage is defined as $A = r - V$, with $r$ a reward and $V$ some tunable baseline. Intuitively, lowering $V$ emphasizes high-reward samples, while raising it penalizes low-reward ones more heavily. We first provide a theoretical analysis of this off-policy REINFORCE algorithm, showing that when the baseline $V$ lower-bounds the expected reward, the algorithm enjoys a policy improvement guarantee. Our analysis reveals that while on-policy updates can safely leverage both positive and negative signals, off-policy updates benefit from focusing more on positive rewards than on negative ones. We validate our findings experimentally in a controlled stochastic bandit setting and through fine-tuning state-of-the-art LLMs on reasoning tasks.

## 1 Introduction

Reinforcement Learning (RL) has long been applied to align Large Language Models (LLMs) to users' preferences using human feedback [Christiano et al., 2017, Ouyang et al., 2022, Dubey et al., 2024]; more recently, it has been used to augment models in more general ways, and in particular to develop their reasoning, coding and tool use capacities [Shao et al., 2024, Guo et al., 2025, Meta, 2025, OpenAI, 2025]. As RL theoretically allows models to learn beyond the limits of existing training data (see e.g. Silver et al. [2016], AlphaEvolve-team [2025]), we can expect it to play an increasingly crucial role in years to come as models begin pushing scientific boundaries. So far, on-policy techniques, in which models are trained on samples that they generated, have been preferred to off-policy methods, in which models are trained using samples generated from another source, which can e.g. be an outdated version of themselves [Tang et al., 2024]. However, strictly on-policy RL can be difficult or even impossible to implement in many use cases; it also suffers from sample inefficiency, as each trajectory generated cannot be used for more than a single gradient update. This makes some degree of off-policyness unavoidable, and in fact desirable.

As such, we are interested in the study of off-policy RL techniques, with LLMs finetuning as our motivating example. Various sophisticated methods exist that deal with off-policyness; while classical solutions based on Q-learning or value functions often struggle in the context of language modeling

39th Conference on Neural Information Processing Systems (NeurIPS 2025).

[Zheng et al., 2023], losses based on importance sampling correction have yielded good results, though the variance of the importance ratio can be problematic [Precup et al., 2001, Schulman et al., 2017]. In this paper, we consider a simple alternative and study the behavior of a gradient ascent on the expected objective

$$J(\pi) = \mathbb{E}_{y \sim \mu} \left[ \log \pi(y)(r(y) - V) \right] \tag{1}$$

as a function of the baseline $V \in \mathbb{R}$, where $\mu$ and $\pi$ are the behavior (sampling) and current policies respectively and $r(y)$ is the reward of trajectory $y$. We call this algorithm *Asymmetric REINFORCE (AsymRE)*. It is asymmetric in the following sense: higher values for $V$ put more emphasis on pushing down the probability of trajectories with low rewards (i.e. "failures") while mainly ignoring high rewards (i.e., "successes"), whereas a small $V$ results in a more positive approach, where the model mostly increases the probability of successful trajectories while mainly ignoring failures. Our core intuition is that while a model can learn from both its successes and failures while training on-policy, it has less to learn from the failures of another model (i.e., when off-policy), which it might not be likely to produce in the first place, and as such should focus more heavily on the positive examples while training off-policy.

Our key contributions are as follows.

- In Theorem 4.2, we show that the AsymRE algorithm applied in a tabular setting converges to a limit policy $\pi^*_{\mu,V}$ which we characterize. We show that when the baseline $V$ is smaller than the expected reward $V^\mu$ of the behavior policy $\mu$, the limit policy $\pi^*_{\mu,V}$ improves upon the behavior policy $\mu$, while maintaining a wide support. When $V$ becomes larger than $V^\mu$, a phase transition occurs and the support of $\pi^*_{\mu,V}$ shrinks dramatically.

- In Theorem 4.3, we study the iterative application of AsymRE in a policy improvement scheme when $V < V^\mu$. In combination with Theorem 4.2, we see that letting $V \geq V^\mu$ results in premature convergence to a potentially suboptimal policy.

- We then verify our findings in a controlled yet rich setting, that of multi-armed bandits.

- Finally, we validate our results on a larger scale by training Llama 8B and Qwen 3B models on real-world data with AsymRE.

These results confirm our initial intuition that in an off-policy setting, conservatively picking a small baseline is the correct strategy.

## 2 Related works

**Reinforcement learning for LLMs.** Reinforcement learning methods are rapidly becoming the dominant paradigm for fine-tuning LLMs on complex tasks, such as mathematical reasoning and coding. It benefits from certain key advantages over supervised training methods: e.g., models can generate their own training samples in the absence of preexisting data of high enough quality. Reinforcement Learning from Human Feedback (RLHF) has emerged as a cornerstone methodology for aligning large language models with human values and preferences [Achiam et al., 2023]. Early systems [Ouyang et al., 2022] turn human preference data into reward modeling to optimize model behavior accordingly. DPO [Rafailov et al., 2023] has been proposed as a more efficient approach that directly trains LLMs on preference data. However, as LLMs evolve during training, continuing training on pre-generated preference data becomes suboptimal due to the distribution shift from off-policy data. Thus, the need arises for additional data to be collected during mid-training—a key phase in the iterative fine-tuning of LLMs [Touvron et al., 2023, Bai et al., 2022, Xiong et al., 2024, Guo et al., 2024]. One line of works aims to mitigate by merging on-policy and off-policy data to achieve improved performance or alignment [Gu et al., 2017, Feng et al., 2025].

In verifiable domains, recent methods like GRPO [Shao et al., 2024, Guo et al., 2025] have shown strong performance by leveraging binary reward signals. GRPO is a REINFORCE-style algorithm [Williams, 1992] that incorporates negative examples, increasingly recognized as an important ingredient for efficient learning [Ahmadian et al., 2024]. At its core, REINFORCE is an on-policy algorithm, with theoretical guarantees only when the reference model matches the trained model, limiting the user's ability to reuse past data. However, training is rarely truly on-policy: data is generated in a parallel, asynchronous way. Moreover, given the computational cost of rollouts, reusing of trajectories is often desirable.

**Off-policy approaches.** While off-policiness is well-studied in RL [Degris et al., 2012], it is not the case in the context of LLMs. Algorithms like REINFORCE tend to become unstable with off-policy data [Pang and He, 2021]. One way to mitigate these issues is to introduce KL-regularization towards the data-generating policy or early stopping, which effectively diminish the influence of negative trajectory and the amount of off-policy learning. Q-learning [Watkins and Dayan, 1992] is also able to handle off-policy data but is less suitable for deployment in LLMs. Perhaps the most common technique to address off-policy distribution shift is importance sampling in conjunction with clipping in REINFORCE-style algorithms [Schulman et al., 2017, Munos et al., 2016, Espeholt et al., 2018]. In practice, though, it is well-known that importance sampling is plagued with excessive variance [Precup et al., 2001]. Other approaches to off-policy RL use a consistency condition derived from the explicit solution of the KL-regularized policy optimization problem, which could be enforced on any data, off- or on-policy, see e.g., Rafailov et al. [2023], Richemond et al. [2024], Tang et al. [2025], Cohen et al. [2025]. Concurrent work to ours [Roux et al., 2025] introduces an asymmetric variant of importance sampling to speed up learning, noting in passing that the baseline $V$ in REINFORCE plays a role in connection with negative samples in off-policy data. Zhu et al. [2025] also investigates the importance of negative and positive samples, though in an on-policy setting.

## 3 Setting

In reinforcement learning (RL), the task is to maximize the expected reward

$$V^\pi \stackrel{\text{def}}{=} \mathbb{E}_{y\sim\pi}\left[r(y)\right] \tag{2}$$

of a trainable *current policy* $\pi$ given some reward function $r$. This is typically done by iteratively updating $\pi$ using training samples $(y, r(y))$, where the trajectories $y$ are sampled from some *behavior policy* $\mu$; here, each trajectory $y$ potentially represents a whole sequence of states (or observations) and actions taken according to the policy, with partial rewards reflected in $r(y)$. If the training samples $y$ are drawn from the current policy, i.e. if $\mu$ is kept equal to the evolving policy $\pi$, it is called *on-policy* RL; otherwise, it is *off-policy* RL.

In the absence of a superior behavior policy $\mu$ from which to sample, algorithms that are as on-policy as possible, i.e. in which $\mu$ is kept as close to the trained policy $\pi$ as possible by frequently updating it, is usually preferable for LLM training [Tang et al., 2024]. However, on-policy methods suffer from various limitations. Though conceptually simple, they pose delicate engineering problems due to the need for asynchronous cooperation among multiple GPUs, often leading to suboptimal resource utilization. Furthermore, for $\mu$ to remain equal to $\pi$, the algorithm must wait for the current batch of trajectories to be generated and evaluated using the reward function $r$, and for $\pi$ (and $\mu$) to be updated, before a new batch can be produced. In many common settings, such as code generation, agentic interaction with the web or human feedback, rewards can take minutes or even hours to be produced; this is even more true for embedded agents interacting with the physical world. Fully on-policy methods can become extremely inefficient or even infeasible due to this. Finally, a strictly on-policy algorithm can only use the trajectories it generates for a single policy update before having to discard them, whereas one might want to reuse samples generated earlier during training, or even coming from the training of another model. This makes them very sample-inefficient, and consequently compute-inefficient, as trajectories are often costly to generate.

This makes off-policy algorithms an attractive alternative [Rafailov et al., 2023, Richemond et al., 2024, Tang et al., 2025, Cohen et al., 2025]; we are in particular interested in off-policy RL with *delayed updates*, in which the actor policy $\mu$ is simply an outdated version of the current policy $\pi$, and is updated (by setting it equal to $\pi$) every $N$ training iterations of $\pi$. Common off-policy algorithms typically apply importance sampling correction to obtain unbiased estimates of the objective function [Schulman et al., 2017, Espeholt et al., 2018, Tang et al., 2025]. Both approaches have drawbacks; in this paper, we investigate a third method for handling off-policy data.

**Context-dependent tasks and LLMs.** Context-dependent tasks, such as the ones for which LLMs are designed, add another layer of subtlety to the study of RL algorithms. In the setting above, we considered a single objective function, namely the expected reward from Equation 2. Conversely, the behavior policy $\pi$ of an LLM typically has to generate responses $y \sim \pi(\cdot|x)$ for various prompts $x \sim \rho$ sampled from some task distribution $\rho$, and its goal is to maximize the expected context-dependent reward $\max_\pi \mathbb{E}_{x\sim\rho, y\sim\pi}\left[r(x, y)\right]$. If each prompt $x$ defined an entirely distinct problem,

then whatever RL algorithm we choose to apply would only see a handful of trajectories associated to this problem, and learning would be essentially impossible. This is not so, as the regularities of language and of LLMs' architectures make it so that similar prompts, such as "Please solve 2 + 3 = ?" and "Compute 2 + 3 = ?", yield similar problems and conditional distributions, and training on one question helps with the other. This additional structure on the space of conditioning prompts plays a key role in enabling LLM training, yet is rarely discussed in RL literature.

# 4   Asymmetric REINFORCE

We set ourselves in the general RL setting described earlier in which the goal is to maximize the expected reward $V^\pi = \mathbb{E}_{y \sim \pi}[r(y)]$ of our current policy $\pi$. For that purpose, one may use a simple on-policy *Policy Gradient* (PG) algorithm, REINFORCE Williams [1992], which updates the parameters of the policy in the direction

$$\mathbb{E}_{y \sim \pi}\left[\nabla \log \pi(y)\left(r(y) - V\right)\right], \tag{3}$$

where $V \in \mathbb{R}$ is a value baseline and the gradient is taken with respect to some differentiable parameters. In practice, $V$ is often set to be some learnt approximation $V \approx V^\pi$ of the value function of the current policy, or to be the empirical average $V = \frac{1}{n}\sum_{i=1}^{n} r(y_i)$ of the rewards obtained when generating several responses $y_i$ (see e.g., Ramesh et al. [2024], Shao et al. [2024]).

In the on-policy setting, the role of the value baseline is simply to offer possible variance reduction of the PG update: the expected behavior of the algorithm during the learning process, as well as its asymptotic performance, are not affected by the choice of this baseline. On the contrary, in the off-policy case, the choice of the baseline impacts the expected behavior of both the algorithm and the asymptotic policy. To analyze this behavior, we consider the simplest policy gradient algorithm, namely REINFORCE, applied to the off-policy setting. We call this algorithm **Asymmetric REINFORCE** (or **AsymRE** in short):

**Definition 4.1** (Expected AsymRE and AsymRE). *Given a behavior policy $\mu$ and a baseline $V \in \mathbb{R}$, the* expected Asymmetric REINFORCE *is a gradient ascent in the direction*

$$\mathbb{E}_{y \sim \mu}\left[\nabla \log \pi(y)\left(r(y) - V\right)\right],$$

*The* Asymmetric REINFORCE *is its stochastic counterpart using samples drawn from $\mu$.*

This corresponds to a stochastic gradient ascent on the expected objective:

$$J(\pi) = \mathbb{E}_{y \sim \mu}\left[\log \pi(y)\left(r(y) - V\right)\right]. \tag{1}$$

Note that it does not coincide with the expected reward of $\pi$, which is our true objective. As it does not involve any off-policy correction, such as importance sampling, we do not expect this algorithm to converge to an optimal policy in general. However we will see that even off-policy this algorithm offers policy improvement guarantees (under some condition on the choice of the baseline $V$).

**Intuition**   The baseline $V$ can be understood as a way to control the emphasis put on the good trajectories (those with high rewards) versus the bad ones (low rewards). If $V$ is large, greater (absolute) weight is given to gradient updates from bad trajectories, which are pushed down, and vice versa, if $V$ is low, more weight is given to push-up the good ones. This matters not in an on-policy setting, since the baseline does not introduce bias to the expected dynamics. However, in the off-policy setting, the choice of the baseline in AsymRE changes the expected behavior of the policy gradient and provides us with a way to introduce some asymmetry in our treatment of good versus bad trajectories. In the simplest of settings, in which rewards are binary $r(y) \in \{0, 1\}$, letting $V = 0$ is equivalent to applying supervised-learning on good trajectories while ignoring the bad ones. Conversely, setting $V = 1$ means learning from mistakes only, while ignoring good rewards. Intuitively, we expect a comparatively lower baseline to be more beneficial in an off-policy setting, as the failures of another policy are less informative for the current policy; we explore this intuition in the remainder of this article.

**Dynamics of AsymRE in a tabular setting.**   Our first result characterizes the dynamics and limits of the AsymRE algorithm in the case of a tabular softmax policy representation.

**Theorem 4.2.** *[Analysis of expected AsymRE for tabular softmax policies] Let $Y$ be a finite set, $\mu$ be some behavior policy whose support is $Y$, and consider a softmax policy representation $\pi(y) \stackrel{def}{=} e^{l(y)} / \sum_{y'} e^{l(y')}$ on $Y$, where the logits $\{l(y)\}_{y \in Y}$ are the policy parameters. We consider the expected AsymRE algorithm with respect to the logits initialized at some $\pi_0$ with successive iterates $\pi_{\mu,V}^t$ and learning rate $\eta$ as per Definition 4.1. Then the AsymRE algorithm converges to a limit distribution $\pi_{\mu,V}^*$. The baseline parameter $V$ controls the nature of the limit distribution:*

- *If $V < V^\mu$, then $\pi_{\mu,V}^*$ is defined by*

$$\pi_{\mu,V}^*(y) = \frac{(\mu(y)(r(y) - V) - \tau_{\mu,V})^+}{V^\mu - V},$$

*where $\tau_{\mu,V}$ is uniquely characterized by the constraint $\sum_{y \in \mathcal{Y}} (\mu(y)(r(y) - V) - \tau_{\mu,V})^+ = V^\mu - V$, and $x^+ = \max(x, 0)$.*

- *If $V = V^\mu$, then $\pi_{\mu,V}^*$ is defined by its support*

$$\text{supp}(\pi_{\mu,V}^*) = \arg\max_{y \in Y} \mu(y)(r(y) - V),$$

*and by $\pi_{\mu,V}^*(y) / \pi_{\mu,V}^*(z) = \pi_0(y)/\pi_0(z)$ for any $y, z \in \text{supp}(\pi_{\mu,V}^*)$.*

- *If $V > V^\mu$, then $\pi_{\mu,V}^*$ can charge any of the elements in the set*

$$\big\{ y \,\big|\, \min_{z \in Y} \mu(y)(r(y) - V) - \mu(z)(r(z) - V) + V - V^\mu > 0 \big\},$$

*depending on the initial condition $\pi_0$.*

*As a consequence, $\text{supp}(\pi_{\mu,V_1}^*) \subseteq \text{supp}(\pi_{\mu,V_2}^*)$ when $V_2 \leq V_1 \leq V^\mu$.*

The proof of this theorem can be found in Appendix A.

**Comparison to on-policy REINFORCE:** The result in Theorem 4.2 is somewhat surprising in the fact that its behavior differs greatly from the on-policy REINFORCE (3). The stable points of the on-policy REINFORCE algorithm are the optimal policies, i.e. those whose support is included in $\arg\max_y r(y)$, and as mentioned above, are independent of the choice of the baseline $V$. However, in the off-policy REINFORCE algorithm, the role played by the baseline $V$ is crucial. Not only does it affect training dynamics, but also the final policies. Indeed, the size of the support of the limit policies $\pi_{\mu,V}^*$ of AsymRE decreases as $V$ increases, while for $V = \min_y r(y)$ the support is $Y$ itself. Thus, the higher $V$, the more deterministic the resulting policies. Moreover, there is an abrupt change of behavior as $V$ becomes equal to or larger than $V^\mu$: at this tipping point, the support suddenly shrinks, and generically becomes a singleton. This phase transition has important consequences which we detail below.

**Policy improvement scheme with AsymRE:** Let $V < V^\mu$, and let us define $\mathcal{T}_V$ the operator that takes as input a behavior policy and returns the limit policy of the AsymRE algorithm (as defined in Theorem 4.2), i.e.:

$$(\mathcal{T}_V \mu)(y) = \pi_{\mu,V}^*(y) \propto (\mu(y)(r(y) - V) - \tau_{\mu,V})^+ \quad \text{for all} \quad y \in Y.$$

Note that this limit policy does not depend on the choice of initial policy $\pi_0$. We now consider the repeated application of $\mathcal{T}_V$ in a policy improvement scheme, in which the behavior policy $\mu_{t+1}$ of each iteration is the limit policy $\mathcal{T}_V \mu_t$ of the previous iteration. The expected dynamics of this process are detailed in the following theorem, whose proof can be found in Appendix A.

**Theorem 4.3.** *[Policy improvement dynamics] Let $\mu$ be any policy with support $Y$, and let $V < V^\mu$.*

1. *Each application of the AsymRE algorithm increases the expected reward: $V^{\mathcal{T}_V \mu} \geq V^\mu$.*

2. *The sequence of expected rewards $V^{(\mathcal{T}_V)^n \mu}$ converges to some limit expected reward $V^\infty$. Let $Y^\infty \stackrel{def}{=} \{y \,:\, r(y) = V^\infty\}$. Then the mass of $(\mathcal{T}_V)^n \mu$ concentrates exponentially fast on $Y^\infty$, i.e.*

$$\sum_{y \notin Y^\infty} ((\mathcal{T}_V)^n \mu)(y) \leq c^n$$

*for some $c < 1$.*

3. *There exists $V_{0,\mu}$ such that the corresponding limit reward is optimal (i.e. $V^\infty = \max_{y \in Y} r(y)$) if and only if $V < V_{0,\mu}$.*

In practice, the number of training steps in each iteration of a policy improvement scheme is finite; the more steps there are, the more off-policy the training. As the number of steps within an iteration increases, the current policy gets closer to the limit policy, whose support is always smaller than that of the initial policy, and smaller the larger $V$ is. This can lead to the premature elimination of points $y$ with high reward. Hence the more off-policy the training, the smaller the baseline needs to be to prevent premature convergence to a suboptimal solution. Interestingly, it can be shown that one can be over-conservative: letting $V$ be very small results in much slower convergence rates. Thus there is a trade-off between the speed of convergence and the risk of reaching a suboptimal limit policy.

Note that if $V \geq V^\mu$, then the first point of the theorem does not hold any more and $\mathbb{E}_{y \sim \pi^*_{\mu,V}}[r(y)]$ can be smaller than $\mathbb{E}_{y \sim \mu}[r(y)]$. Furthermore, the support of the policy shrinks at the first policy improvement iteration (as described in Theorem 4.2), which makes all subsequent iterations useless.

**From the tabular setting to LLMs**  As explained at the end of Section 3, LLMs are applied to a variety of conditioning contexts $x$ (the prompts), each of which corresponding to a distinct RL problem. The straightforward adaption of the AsymRE method to this setting is to train an LLM's policy $\pi$ using the context-averaged version of our loss

$$\mathbb{E}_{x \sim \mathcal{D}, y \sim \mu(\cdot|x)} \left[\log \pi(y|x) \left(r(y,x) - V\right)\right],$$

where $\mathcal{D}$ is some distribution of prompts. Translating our theoretical findings to the case of LLMs requires some cautiousness. Note first that the expected reward $V^{\mu(\cdot|x)} = \mathbb{E}_{y \sim \mu(\cdot|x)}[r(y,x)]$, whose value relative to $V$ conditions the behavior of the AsymRE algorithm, differs for each prompt. A natural solution is to consider instead the context-corrected loss

$$\mathbb{E}_{x \sim \mathcal{D}, y \sim \mu(\cdot|x)} \left[\log \pi(y|x) \left(r(y,x) - V^{\mu(\cdot|x)} - \delta V\right)\right], \tag{4}$$

so that the critical value becomes $\delta V = 0$ uniformly across the $x$s. Furthermore, though our theorems apply to each of the problems corresponding to the various prompts, the averaged behavior of the model over the distribution of problems is more subtle. At last, our results focus on the training loss of the trained policy, whereas we might be interested in the LLM's test loss on contexts on which they were not trained.

Nonetheless, we can hope that the regularities of LLMs with respect to prompts can allow us to partially predict the effects of training on certain prompts on the LLM's policy conditioned by other prompts. In particular, our results suggest that letting $V$ be equal to or larger than 0 in the context-corrected loss in Equation 4 can lead to dramatic decrease in the diversity of answers, corresponding to the collapse of the policy's support predicted by our theorems. While this may not necessarily penalize the expected reward of a tabular policy, it is detrimental to a language model; in particular, it poses a risk of severe overfitting of the training set, and consequently of poor test results.

## 5 Experiments

We validate our findings first in a controlled setting, then with LLMs on real-world data.

### 5.1 Bandits

We illustrate our findings in a simple bandit setting. There are 100 arms. The expected reward $r(y)$ of each arm $y$ is chosen uniformly randomly in $[0, 1]$. We sample from a non-uniform behavior policy $\mu$ defined as a softmax of the logits $l(y) = y/10$, where $y$ is indexed from 0 to 99. The current policy $\pi_t$ is a softmax of its logits $l_t(y)$, initialized as $\pi_0 = \mu$, and its logits are updated according to the expected AsymRE algorithm with learning rate 1:

$$l_{t+1} = l_t + \nabla_l \mathbb{E}_{y \sim \mu} \left[\nabla \log \pi_t(y) \left(r(y) - V\right)\right].$$

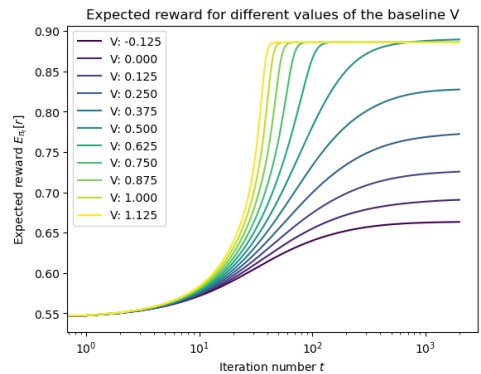

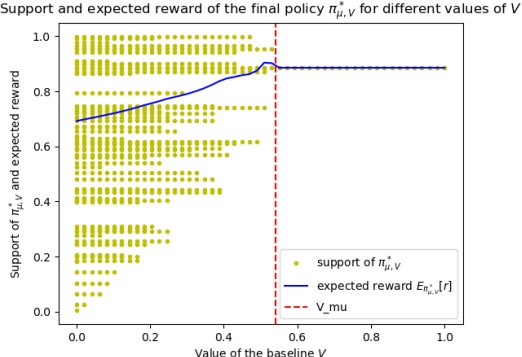

(a) Expected reward $\mathbb{E}_{y \sim \pi_t}[r(y)]$ of the AsymRE algorithm as a function of the training iteration $t$ and baseline $V$.

(b) The support of the final policy $\pi_{\mu,V}^*$ is shown by the yellow dots; each dot corresponds to an arm (sorted by increasing reward values). The support suddenly drops to a single atom when $V$ crosses $V^\mu \approx 0.54$.

Figure 1: Expected rewards and supports of the policies in the bandits experiments

**Expected reward and support with AsymRE** In Figures 1a and 1b, we report the expected reward $\mathbb{E}_{y \sim \pi_t}[r(y)]$ of the expected AsymRE algorithm. We observe that the performance of the algorithm tends to improve as the value of the baseline $V$ gets closer to some $V_{0,\mu} < V^\mu \approx 0.5405$. However, the performance of all policies is upper bounded by a sub-optimal value $\approx 0.89$, which is lower than the optimal expected reward ($\max_y r(y) = 0.999$ for this run).

Although these experiments may seem to support the choice of selecting a high value of the baseline, Figure 1b, where we plot the support of the final policy $\pi_{\mu,V}^*$ as a function of $V$, shows a complete loss of diversity in the resulting policies for high values of $V$, making future improvements in a policy iteration scheme impossible. This is also supported by the drastic decrease in entropy for high values of $V$ reported in Figure 7, in the Appendix. We observe a phase transition for $V$ around $V^\mu$. For $V < V^\mu$, the size of the support decreases as $V$ increases, as predicted in the Theory Section 4. The policy stays supported by a large number of atoms even when $V \approx V^\mu$, as long as $V < V^\mu$. But as soon as $V \geq V^\mu$, the support of the optimal policy $\pi_{\mu,V}^*$ becomes a singleton: $\arg\max_y \mu(y)(r(y) - V)$.

Thus, when $V \geq V^\mu$, there is a risk that our off-policy REINFORCE algorithm might produce policies that lose their diversity. In addition, notice that when $V$ is slightly larger than $V^\mu$, the reward of the corresponding limit policy is not as large as the reward of the best arm in the support of the limit policy for $V$ slightly smaller than $V^\mu$. Thus the potential of finding this superior arm is lost forever when $V$ crosses $V^\mu$.

**Policy improvement with AsymRE** We study a policy improvement scheme over $40$ iterations, where at each iteration we apply the AsymRE algorithm for $500$ steps using the policy obtained at the end of the previous iteration as both our behavior policy and starting current policy. In Figure 2, we show the expected reward of the corresponding policies. We notice that when $V < V^\mu$, for every policy iteration step, the performance of the improved policies $\mathcal{T}_V \mu_t$ are better than the corresponding behavior policies $\mu_t$, as predicted by Theorem 4.3. However, for $V \geq V^\mu$, we have already seen in Figure 1a that the policy converges to a deterministic and suboptimal policy. Here we see that iterating this algorithm does not permit to recover from this premature suboptimal convergence, regardless of the value of $V \geq V^\mu$. We also observe that while they do not plateau as dramatically, trajectories corresponding to $V$ only slightly smaller than $V^\mu$ (such as $0.525$ and $0.54$) converge to a less favorable limit policy than for $V \leq 0.5$.

## 5.2 Large Language Models

We validate our findings by training LLMs on reasoning tasks.

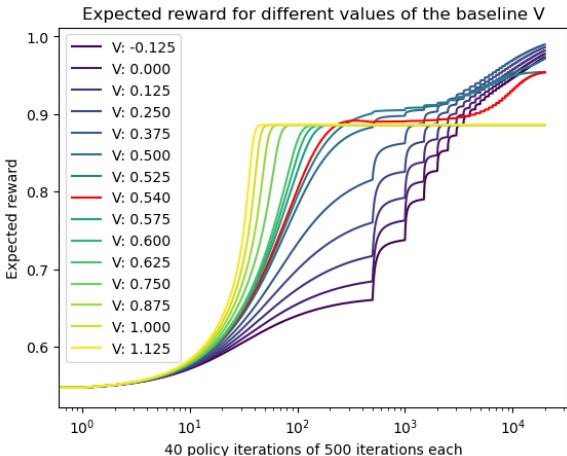

Figure 2: Performance of the current policies for 40 iterations of policy improvement with AsymRE. Each iteration is 500 steps. The curve corresponding to $V \approx V^{\mu}$ is in red.

**Experimental setup**  We train Llama-3.1-8B-Instruct [Dubey et al., 2024] (which we refer to as Llama 8B), Llama-3.2-3B-Instruct (which we refer to as Llama 3B) and Qwen2.5-3B-Instruct [Yang et al., 2024] (which we refer to as Qwen 3B) with the AsymRE objective

$$J(\pi) = \mathbb{E}_{x \sim \mathcal{D}, \{y_i\}_{i=1}^{G} \sim \mu(.|x)} \left[ \frac{1}{G} \sum_{i=1}^{G} (r(y_i, x) - (\hat{V} + \delta V)) \log(\pi(y_i|x)) \right],$$

where $x$s are prompts from the dataset $\mathcal{D}$, each $y_i$ is a full sequence of tokens auto-regressively sampled from the LLM $\mu$, and $\hat{V} = \frac{1}{G} \sum_{i=1}^{G} r(y_i, x)$ is the empirical estimate of the average reward $V^{\mu(\cdot|x)}$. This loss is the empirical approximation of the context-corrected loss in Equation (4). We place ourselves in the delayed updates setting, i.e. the behavior policy $\mu$ is simply an outdated version of the current policy $\pi$, which we update every $N$ training steps; this corresponds to a policy improvement scheme where each iteration lasts $N$ steps. We set the number $G$ of samples per prompt to $8$. We train and test on the MATH dataset (Hendrycks et al. [2021], 12.5k high school-level problems), as well as on a subset of size 142k of the NuminaMath dataset (LI et al. [2024], competition-level problems). The reward of a trajectory is $1$ if the answer is correct, and $-1$ otherwise. Additional details regarding the experimental setup are given in Appendix B.

**Impact of the baseline $\delta V$**  The train and test losses for various baselines $\hat{V} + \delta V$ are shown in Figures 3 and 4 (Llama 8B and Qwen 3B respectively). Similarly as what we observe with the bandits experiment (Figure 2), while the baseline $\delta V$ is smaller than the critical value $0$, the model's training accuracy tends to increase with $\delta V$. This is reflected in the test accuracy as well, though less strongly. When the baseline $\delta V$ reaches, then passes $0$, a phase transition occurs and both the training loss and the test loss suffer a catastrophic collapse. The larger $\delta V \geq 0$, the faster this collapse occurs (for $\delta V = 0$, the collapse occurs at the very end of our training run). This corresponds in the theory (and in the bandit setting) to the regime where the limit policy becomes deterministic: while this can result in a good expected reward in the single-problem bandit setting, it leads to a drastic decrease in accuracy in the LLM setting, where the model must retain diversity to answer the range of problems with which it is tasked. This loss of diversity can be observed in the entropy of the models, in Figure 8 in the Appendix.

The left side of Figure 5 confirms our claim that letting $\delta V$ be slightly smaller than $0$ results in greater training stability: we compare 7 independent training runs with $\delta V = 0$ and 7 training runs with $\delta V = -0.1$, and we find that all runs with $\delta V = 0$ collapse.

**Comparison to GRPO**  Though further testing would be needed to reach definitive conclusions, we also observe that our loss compares favorably to GRPO both in terms of training speed and final accuracy in the off-policy setting, as shown on the right side of Figure 5.

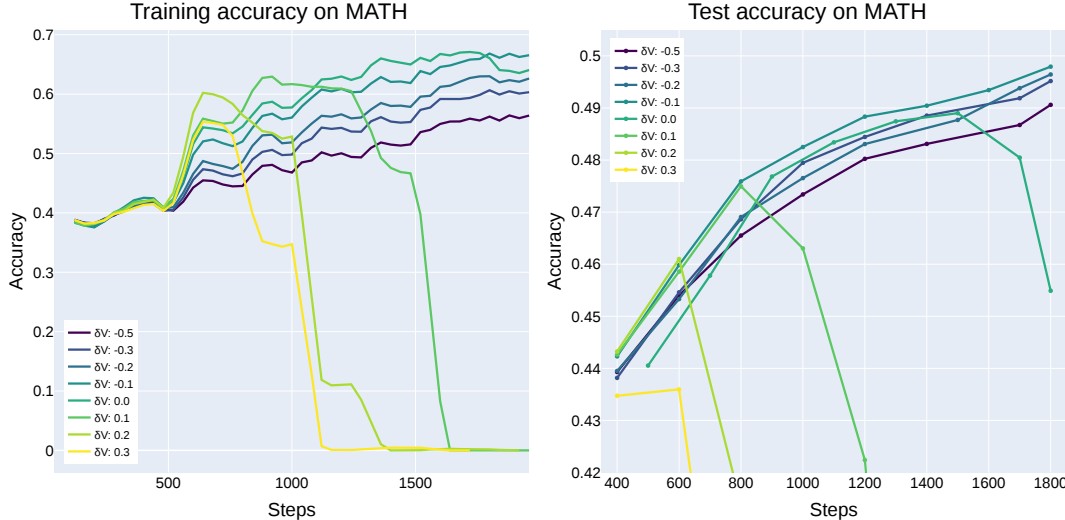

Figure 3: Training dynamics of Llama 8B on the MATH dataset (results are averaged over 3 seeds, and a moving average with a window of size 3 is applied). The behavior policy is updated every $N = 250$ training steps.

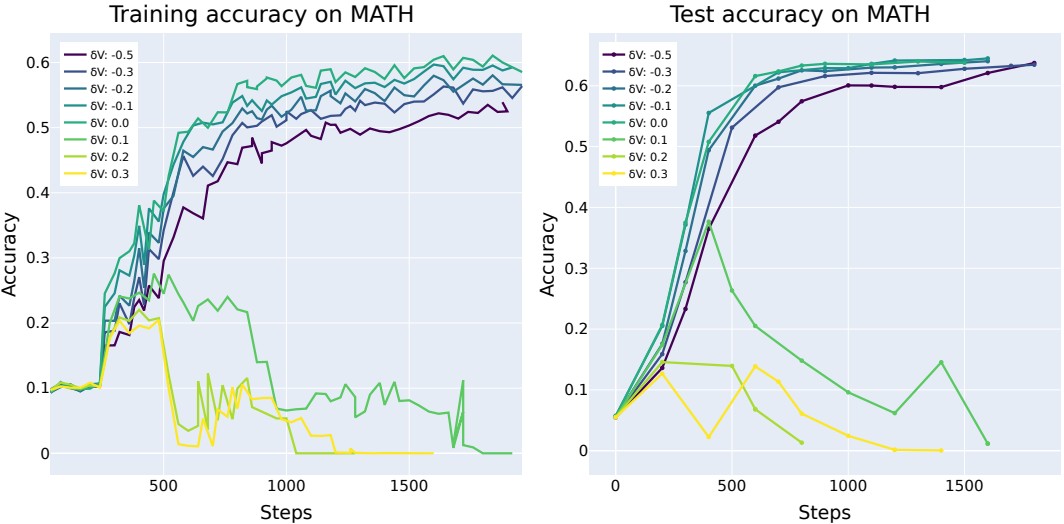

Figure 4: Training dynamics of Qwen 3B on the MATH dataset (results are averaged over 3 seeds, and a moving average with a window of size 3 is applied). The behavior policy is updated every $N = 250$ training steps.

Our additional results in Appendix C on the larger NuminaMath dataset, as well as on Llama 3B, support the same conclusions.

## 5.3 Discussion

These experiments corroborate our theoretical findings and can be summarized as follows:

- Letting the baseline $V$ be strictly smaller than the expected reward $V^\mu$ of the behavior policy $\mu$ makes the training stable, even when off-policy, and offers a monotonic improvement of the policies.

- Performance tends to improve as $V$ grows closer to $V^\mu$ while remaining strictly smaller.

- As soon as $V$ passes $V^\mu$, the policies become more deterministic. In the context of bandits, this leads to a suboptimal convergence of the policy improvement scheme. In the context of LLMs, this leads to a collapse of the training and test loss.

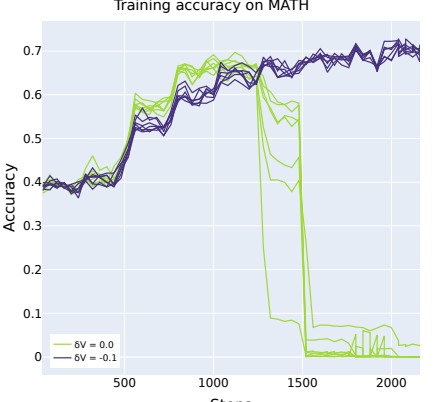
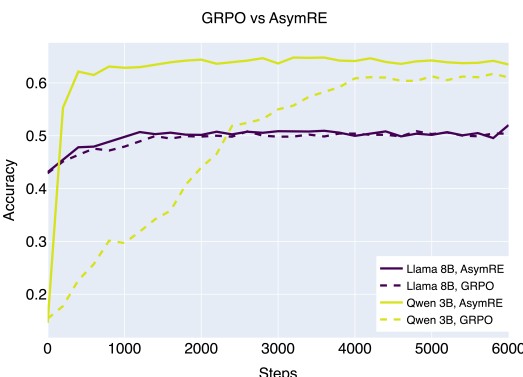

Figure 5: *Left:* Training dynamics of Llama 8B on the MATH dataset for two values of the baseline, $\delta V = 0$ and $\delta V = -0.1$, and 7 independent runs for each value. The behavior policy is updated every $N = 250$ steps. We observe a systematic collapse when $\delta V = 0$. *Right:* Test accuracy of Llama 8B and Qwen 3B trained on the MATH dataset with GRPO and AsymRE (with $V = -0.1$). The behavior policy is updated every $N = 250$ steps. Asymmetric REINFORCE leads to faster convergence and better than GRPO.

While further large scale experiments are needed to draw definitive conclusions regarding the training of LLMs, our findings suggest that adding a small conservative correction $\delta V \approx -0.1$ in the advantage term of the training objective

$$\mathbb{E}_{x \sim \mathcal{D}, \{y_i\}_{i=1}^{G} \sim \mu(.|x)} \Big[ \frac{1}{G} \sum_{i=1}^{G} (r(y_i, x) - (\hat{V} + \delta V)) \log(\pi(y_i|x)) \Big]$$

might result in greater training stability, and help prevent the collapses often observed in RL training.

## 6   Conclusion

We have thoroughly studied the behavior of our simple off-policy algorithm AsymRE with respect to the baseline $V$, both theoretically and experimentally. We have confirmed our claims that in an off-policy setting, selecting a baseline slightly smaller than the expected reward of the behavior policy results in superior asymptotic performances, and can help alleviate the risk of training collapse. Such a choice of baseline corresponds to putting more emphasis on positive training examples, and less on negative ones, which is intuitively reasonable.

**Limitations and possible improvements**   While we have analyzed in depth the role of the baseline $V$ in our simple AsymRE method, it would be interesting to extend our study to more sophisticated losses, which might e.g. include importance ratio correction and KL-divergence regularization. Finally, the possibility to leverage off-policyness by training over the same samples for multiple epochs was one of our initial motivations: quantifying the potential run-time and compute gains and the impact on model performance of such a scheme is an exciting future research direction.

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

# A Proofs of Theorems 4.2 and 4.3

## A.1 Proof of Theorem 4.2

We first prove Theorem 4.2, which we restate for the reader's convenience:

**Theorem 4.2.** *[Analysis of expected AsymRE for tabular softmax policies] Let $Y$ be a finite set, $\mu$ be some behavior policy whose support is $Y$, and consider a softmax policy representation $\pi(y) \overset{def}{=} e^{l(y)} / \sum_{y'} e^{l(y')}$ on $Y$, where the logits $\{l(y)\}_{y \in Y}$ are the policy parameters. We consider the expected AsymRE algorithm with respect to the logits initialized at some $\pi_0$ with successive iterates $\pi_{\mu,V}^t$ and learning rate $\eta$ as per Definition 4.1. Then the AsymRE algorithm converges to a limit distribution $\pi_{\mu,V}^*$. The baseline parameter $V$ controls the nature of the limit distribution:*

- *If $V < V^\mu$, then $\pi_{\mu,V}^*$ is defined by*

$$\pi_{\mu,V}^*(y) = \frac{(\mu(y)(r(y) - V) - \tau_{\mu,V})^+}{V^\mu - V},$$

  *where $\tau_{\mu,V}$ is uniquely characterized by the constraint $\sum_{y \in \mathcal{Y}} (\mu(y)(r(y) - V) - \tau_{\mu,V})^+ = V^\mu - V$, and $x^+ = \max(x, 0)$.*

- *If $V = V^\mu$, then $\pi_{\mu,V}^*$ is defined by its support*

$$\mathrm{supp}(\pi_{\mu,V}^*) = \arg\max_{y \in Y} \mu(y)(r(y) - V),$$

  *and by $\pi_{\mu,V}^*(y)/\pi_{\mu,V}^*(z) = \pi_0(y)/\pi_0(z)$ for any $y, z \in \mathrm{supp}(\pi_{\mu,V}^*)$.*

- *If $V > V^\mu$, then $\pi_{\mu,V}^*$ can charge any of the elements in the set*

$$\left\{ y \,\big|\, \min_{z \in Y} \mu(y)(r(y) - V) - \mu(z)(r(z) - V) + V - V^\mu > 0 \right\},$$

  *depending on the initial condition $\pi_0$.*

*As a consequence, $\mathrm{supp}(\pi_{\mu,V_1}^*) \subseteq \mathrm{supp}(\pi_{\mu,V_2}^*)$ when $V_2 \le V_1 \le V^\mu$.*

*Proof.* For simplicity, we omit the indices when the context makes them unnecessary; in particular, we write $\pi_t$ for $\pi_{\mu,V}^t$. We are optimizing

$$\mathbb{E}_{y \sim \mu}[(r(y) - V) \log \pi(y)] \tag{5}$$

with the logits parameterization $\pi(y) \propto \exp(l(y))$, hence gradient flow ascent corresponds to

$$\partial_t l = \eta \nabla F(l) = \eta(a_y - b\pi(y)), \text{ where } F(l) \overset{def}{=} \sum_y a_y l(y) - b \log \sum_z \exp(l(z)) \tag{6}$$

and $a_y \overset{def}{=} \mu(y)(r(y) - V)$, $b \overset{def}{=} \sum a_y = V^\mu - V$, and $\eta > 0$ is some speed parameter. This follows from the definition of $\pi(y) = \exp(l_y) / \sum_z \exp(l_z)$, and of $V^\mu = \sum_y \mu(y)r(y)$.

Eventually, one may also consider gradient ascent with a fixed stepsize $\eta > 0$, leading to the evolution

$$l_{t+1} = l_t + \eta \nabla F(l_t) \tag{7}$$

We will see that, as long as $\eta$ is not too large, dynamics (6) and (7) converge to the same limit.

### A.1.1 Case $V^\mu = V$

When $V^\mu = V$, which is equivalent to $b = 0$, we are maximizing a linear function, which leads to $l_t = l_0 + \eta t a$, where $l_0$ is the initial condition of the logits and $a \overset{def}{=} (a_y)_{y \in Y}$. This implies, assuming $\eta = 1$ without loss of generality,

$$\pi_*(y) = \lim_{t \to \infty} \frac{\exp(a_y t + l_0(y))}{\sum_z \exp(a_z t + l_0(z))} \propto \mathbb{I}\{y \in \arg\max_z a_z\} \exp(l_0(y)) \propto \mathbb{I}\{y \in \arg\max_z a_z\} \pi_0(y).$$

**Speed of convergence.** Note that in this case, we can also characterize the speed of convergence: if $a_y \neq a_* \stackrel{\text{def}}{=} \max a_z$,

$$\pi_t(y) = \frac{\exp(a_y t + l_0(y))}{\sum \exp(a_z t + l_0(z))} \leq \exp((a_y - a_*)t + l_0(y) - \min l_0(z)),$$

which goes to zero exponentially fast as $\exp((a_y - a_*)t)$. If $a_y = \max a_z$, then

$$\pi_t(y) = \frac{\exp(a_y t + l_0(y))}{\sum \exp(a_z t + l_0(z))}$$

$$= \frac{\exp(a_y t + l_0(y))}{\exp(a_y t)\left(\sum_{z \in \arg\max a_y} \exp(l_0(z)) + \sum_{z \notin \arg\max a_y} \exp((a_z - a_y)t + l_0(z))\right)}$$

$$= \pi_*(y) \frac{1}{1 + \frac{\sum_{z \notin \arg\max a_y} \exp((a_z - a_y)t)}{\sum_{z \in \arg\max a_y} \exp(l_0(z))}}$$

Series expansion leads to

$$\|\pi_t - \pi_*\| \leq \frac{\pi_*(y)\,|Y|}{\sum_{z \in \arg\max a_y} \exp(l_0(z))} \exp(\max_{z \notin \arg\max a_y} (a_z - a_y)t)$$

Hence the convergence speed is $A \exp(-ct)$ for $c = \min(a_* - a_y)$ and some constant $A$.

Note that this formula holds for both gradient flow and gradient ascent.

### A.1.2 Case $V \neq V^\mu$

**Convergence of the continuous dynamics.** To show the gradient flow dynamics converge, let us introduce the following function

$$\Phi_t = \sum a_y \pi_t(y) - \frac{b}{2} \sum \pi_t(y)^2.$$

The dynamics on the logits can be cast as a dynamics on the probabilities.

$$\partial_t \pi_t(y) = \partial_t \frac{\exp(l_t(y))}{\sum_z \exp(l_t(z))} = \frac{\exp(l_t(y))\partial_t l_t(y)}{\sum_z \exp(l_t(z))} - \frac{\exp(l_t(y))}{\sum_z \exp(l_t(z))} \frac{\sum_{y'} \exp(l_t(y'))\partial_t l_t(y')}{\sum_z \exp(l_t(z))}$$

$$= \pi_t(y)\partial_t l_t(y) - \pi_t(y) \sum_z \pi_t(z)\partial_t l_t(z)$$

$$= \pi_t(y)(a_y - b\pi(y)) - \pi_t(y) \sum_z \pi_t(z)(a_z - b\pi_t(z))$$

$$= \pi_t(y)(a_y - b\pi_t(y) - \sum_z a_z \pi_t(z) + b \sum_z \pi_t(z)^2) = \pi_t(y)(a_y - b\pi_t(y) - \tau_t),$$

where

$$\tau_t = \sum_z a_z \pi_t(z) - b \sum_z \pi_t(z)^2.$$

Casting this dynamics on $\Phi$, we have

$$\partial_t \Phi_t = \sum_y (a_y - b\pi_t(y))\partial_t \pi_t(y) = \sum_y (a_y - b\pi_t(y))\pi_t(y)(a_y - b\pi_t(y) - \sum_z \pi_t(z)(a_z - b\pi_t(z)))$$

$$= \sum_y \pi_t(y)(a_y - b\pi_t(y))^2 - \sum_y \pi_t(y)(a_y - b\pi_t(y)) \sum_z \pi_t(z)(a_z - b\pi_t(z)))$$

$$= \sum_y \pi_t(y)(a_y - b\pi_t(y))^2 - \left(\sum_y \pi_t(y)(a_y - b\pi_t(y))\right)^2$$

$$= \mathrm{Var}_{Y \sim \pi_t}(a_Y - b\pi_t(Y)) \geq 0,$$

with equality when $a_y - b\pi_t(y)$ is constant on the support of $\pi_t$. Because $V$ is continuous and bounded above on the simplex, which is compact, $\Phi_t$ converges and $\partial_t \Phi_t$ goes to zero. As $V \neq V^\mu$ (which means that $b \neq 0$), this shows that $\pi_t(y)$ either goes to zero, or converges to $(a_y - \tau_*)/b$ for some constant $\tau_*$. Thus, $\pi_t$ converges to some limit distribution $\pi_*$.

Since $\pi_*(y)$ is non-negative, we deduce that, with $Y_* \subseteq Y$ the support of $\pi_*$,

$$\pi_*(y) = \mathbb{I}\{y \in Y_*\} \frac{(a_y - \tau_*)^+}{b}.$$

### A.1.3 Case $V^\mu > V$

In the case $V^\mu > V$, i.e. $b > 0$, we will now show that $Y_*$ is actually defined by

$$Y_* = \{y \,|\, a_y - \tau_* > 0\}.$$

Let us proceed by contradiction. Assume the existence of a $y$ such that $a_y - \tau_* > 0$ and for which $\pi_*(y) = 0$. Then, by continuity of $\tau_t$, there exists an $\varepsilon$ and a $T$ such that for any $t > T$, we have

$$\pi_t(y) \leq \varepsilon/b, \qquad a_y - \tau_t > 2\varepsilon,$$

which implies

$$\partial_t \pi(y) = \pi(y)(a_y - b\pi(y) - \tau_t) > \varepsilon\pi(y).$$

Due to the logits parameterization, we have $\pi(y) > 0$, which leads to $\pi_t(y) \geq A \exp(\varepsilon t)$ for some $A > 0$, which is a contradiction.

Finally, the fact that $\sum \pi_*(y) = 1$ uniquely characterizes $\tau_*$. Indeed, consider the function

$$G : \tau \mapsto \sum \frac{(a_y - \tau)^+}{b}.$$

Then

$$G(\tau_*) = \sum \frac{(a_y - \tau_*)^+}{b} = \sum \pi_*(y) = 1.$$

The function $G$ is continuous and strictly decreasing on $[\min a_y, \max a_y]$, with $G(\max a_y) = 0$ and $G(0) \geq 1$. If $\min a_y < 0$, then $G(0) > 1$, which implies the existence of a unique $\tau_* \in (0, \max a_y]$ such that $G(\tau_*) = 1$. If $\min a_y \geq 0$, then the only solution of $G(\tau_*) = 1$ is $\tau_* = 0$.

**Inclusion of the supports.** From the previous results, we already have the inclusion of the support of $\mathcal{T}_{\mu,V^\mu}$ in that of $\mathcal{T}_{\mu,V}$ for any $V \leq V^\mu$. Let us now consider $V_2 \leq V_1 < V^\mu$, and let $\tau_{\mu,V_1}$, respectively $\tau_{\mu,V_2}$, be the limit $\tau_*$ introduced above for $V_1$, respectively $V_2$. We would like to show that $\operatorname{supp} \mathcal{T}_{\mu,V_1} \subseteq \operatorname{supp} \mathcal{T}_{\mu,V_2}$, which is equivalent to showing that

$$\mu(y)(r(y) - V_2) - \tau_{\mu,V_2} \leq 0 \qquad \Rightarrow \qquad \mu(y)(r(y) - V_1) - \tau_{\mu,V_1} \leq 0.$$

This is a consequence of the fact that $\tau_{\mu,V}$ is increasing with $V$, which we will prove now. Recall that we have

$$\sum \frac{(a_y - \tau_{\mu,V})^+}{b} = 1,$$

hence

$$\sum (\mu(y)(r(y) - V) - \tau_{\mu,V})^+ = V^\mu - V.$$

For $\varepsilon > 0$ small enough, we have

$$\sum (\mu(y)(r(y) - (V + \varepsilon)) - \tau_{\mu,V})^+ \geq \sum (\mu(y)(r(y) - V - \tau_{\mu,V})^+ - \varepsilon$$

$$= V^\mu - (V + \varepsilon) = \sum (\mu(y)(r(y) - (V + \varepsilon)) - \tau_{\mu,V+\varepsilon})^+,$$

hence $\tau_{\mu,V+\varepsilon} \geq \tau_{\mu,V}$, as $\tau \mapsto \sum (\mu(y)(r(y) - (V + \varepsilon)) - \tau)^+$ is decreasing in $\tau$.

**From gradient flow to gradient ascent.** We notice that $F$ is the sum of a linear term and a log-sum-exp term. Since the log-sum-exp term is strictly concave on $1^\perp$ (i.e. on the hyperplane $\sum l_y = \sum l_y(0)$), due to the sign of $b$, we are maximizing a smooth strictly concave function. Gradient ascent on a concave objective can be seen as a discretization of the gradient flow, and is known to follow the same dynamics as long as the step-size is smaller than twice the inverse of the largest absolute eigenvalue of the Hessian [Ryu and Boyd, 2016, page 17]. In our case, the Hessian of $F$ is equal to $-b$ times the Hessian of the log-sum-exp function, which leads to

$$\nabla^2 F(l) = -b \left( \operatorname{diag}(\pi) - \pi\pi^\top \right).$$

A simple calculation leads to the following bound on the operator norm:

$$\left\| \nabla^2 F \right\| \leq b(\|\operatorname{diag}(\pi)\| + \|\pi\pi^\top\|) = b(\pi(y) + \|\pi\|_2^2) \leq 2b.$$

As a consequence, as long as $\eta < 1/b = 1/(V^\mu - V)$, the gradient ascent converges to the same point as the gradient flow.

**Remarks on the speed of convergence.** *When $V < \min r(y)$.* In this case $\tau_* = 0$, and since $a_y > 0$, this implies that $\pi$ has full support. The conservation of $\sum l_t(y)$, which is a consequence of the fact that

$$\partial_t \sum l(y) = \sum a_y - b \sum \pi(y) = b - b = 0,$$

implies that if some logits go to plus infinity, others should go to minus infinity, which is not possible if $\pi(y) \neq 0$ for all $y$. As a consequence, the logits are bounded, and the maximizer of $F$ is achieved on its domain. Moreover, because $F$ is strictly concave on $1^\perp$ and $\mathcal{C}^\infty$, it is strongly concave and smooth on any compact set, which implies exponential convergence of both gradient flow and gradient ascent [Bubeck, 2015, Theorem 3.10].

*When $V^\mu > V > \min r(y)$.* To quantify the speed of convergence, we can use

$$\lim \partial_t l(y) = a_y - b\pi_*(y) = \mathbb{I}\{y \in Y_*\}\tau_* + \mathbb{I}\{y \notin Y_*\}a_y$$

Because $V > \min r(y)$ and $\mu(y) > 0$ for all $y$ (since $\mu$ has full support), this implies $\min a_y < 0$, which implies $\tau_* > 0$ as we have shown above. For any $y \in Y_*$, we have $\lim \partial_t l(y) = \tau_* > 0$, which means that the logit $l(y)$ will go linearly to plus infinity. For any $y \notin Y_*$, by definition $a_y \leq \tau_*$ and $\lim \partial_t l(z) = a_y$, which means that the logit $l(y)$ behaves as $a_y t + o(t)$. As a consequence, for $a_y < \tau_*$, we have

$$\pi_t(y) = \frac{\exp(l_t(y))}{\sum \exp(l_t(z))} = \frac{\exp(a_y t + o(t))}{\exp(\tau_* t) \sum \exp(-(a_z - \tau_*)^- t + o(t)))} \leq \exp((a_y - \tau_*)t + o(t)).$$

In other words, the mass of such points decreases exponentially fast. The examples below show that similar conclusions cannot be drawn regarding other points.

*Example of slow convergence when $a_y = \tau_* > 0$.* In the event $a_y = \tau_*$, the convergence can be slower. For example, one may consider a situation where $l(y) = \exp(\tau_* t)$ and $l(z) = \exp(\tau_* t - \log(t))$, leading to a convergence in $1/t$. This is the case when

$$Y = \{1, 2, 3\}, \quad \mu = (1/3, 1/3, 1/3), \quad r = (9, 3, -6), \quad V = 0, \quad a = (3, 1, -2), \quad b = 2.$$

Then, one can check that $\tau_* = 1$ and $\pi = (1, 0, 0)$. The dynamics on $\pi_t(2)$ can be written as

$$\begin{aligned}
\partial_t \pi(2) &= \pi(2)(a_2 - b\pi(2) - (\sum a_y \pi(y) - b \sum \pi(y)^2)). \\
&= \pi(2)(1 - 2\pi(2) - (3\pi(1) + \pi(2) - 2\pi(3) - 2\pi(1)^2 - 2\pi(2)^2 - 2\pi(3)^2)) \\
&= \pi(2)(1 - 2\pi(2) - (3(1 - \pi(2)) + \pi(2) - 2(1 - \pi(2))^2 - 2\pi(2)^2 + o(\pi(3)))) \\
&= -4\pi(2)^2 + 4\pi(2)^3 + o(\pi(3)).
\end{aligned}$$

From the previous derivation, we know that $\pi(3)$ will go to zero exponentially fast; hence the dynamics will be dominated by $\partial_t \pi(2) \simeq -4\pi(2)^2$, which leads to $\pi(2) \sim 1/4t$, and the slow convergence.

*Example of slow convergence when $a_y = \tau_* = 0$.* Similarly to the previous example of slow convergence, considering

$$Y = \{1, 2\}, \quad \mu = (1/2, 1/2), \quad r = (2, 0), \quad V = 0$$

leads to $\tau_* = 0$, $b = 1$, $\pi = (1, 0)$ and $\pi_t(2) \sim 1/2t$.

### A.1.4 Case $V^\mu < V$

When $V^\mu < V$, which is equivalent to $b < 0$, we are maximizing (*not minimizing*) a strictly convex function on $1^\perp$, which can lead to various limits depending on initial conditions (see after the end of the proof for a short discussion).

Let us first characterize the union of the supports of the potential limits of $\pi$. Using the previous characterization of $\partial_t \pi$, we get

$$\partial_t \log \frac{\pi(y)}{\pi(z)} = \frac{\partial_t \pi(y)}{\pi(y)} - \frac{\partial_t \pi(z)}{\pi(z)} = a_y - b\pi(y) + \tau_t - a_z + b\pi(z) - \tau_t$$
$$= a_y - a_z - b(\pi(y) - \pi(z)).$$

We deduce that for any $(y, z)$, since $\pi(y) - \pi(z) \leq 1$ and $-b > 0$,

$$\frac{\pi_t(y)}{\pi_t(z)} \leq \frac{\pi_0(y)}{\pi_0(z)} \exp((a_y - a_z - b)t).$$

This means that $\min_z a_y - a_z - b < 0$ implies $\pi_*(y) = 0$. Hence $y$ cannot belong to the support of any potential limit. It is easy to show that situation is the same if $\min_z a_y - a_z - b = 0$, as $\pi_0(y) - \pi_0(z) < 1$, hence $a_y - a_z - b(\pi_0(y) - \pi_0(z)) < 0$ and $a_y - a_z - b(\pi(y) - \pi(z))$ remains smaller than some constant $c < 0$ for all $t$. Reciprocally, if $\min_z a_y - a_z - b > 0$, an initial condition $\pi_0$ close to a Dirac on $y$ will lead to $\pi_* = \delta_y$, hence $y$ belongs to the support of some potential limits.

Showing the convergence in the case of the discrete gradient ascent follows from arguments similar to the the concave case treated above. $\square$

As mentioned in the proof, the dynamics of the case $V^\mu < V$ are harder to describe: the gradient flow equation is a case of *replicator equation* [Cressman and Tao, 2014], and mapping the initial condition $\pi_0$ to the final limit $\pi_*$ is not easy. As shown in Figure 6, the regions $\{\pi_0 \mid \pi_* = \delta_y\}$ are not e.g. polytopes.

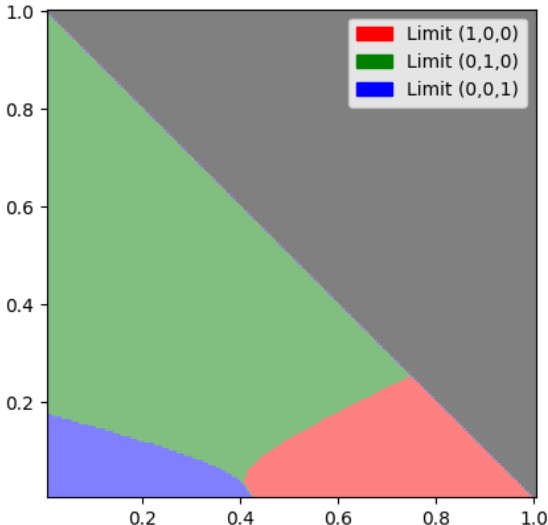

Figure 6: Each point $(x, y)$ in the triangle corresponds to an initial distribution $\pi_0 = (x, y, 1 - x - y$ on a set $Y$ of cardinality $3$. The color of a point $\pi_0$ indicates the limit of a gradient ascent starting with $\pi_0$ as its initial condition: the three colors correspond to $\pi_* = (1, 0, 0)$, $\pi_* = (0, 1, 0)$ and $\pi_* = (0, 0, 1)$ respectively. Note that some initial conditions converge to other limits, but they are of mass $0$ in the simplex and do not appear in the figure.

However, a simple case is when $\arg\max a_y \cap \arg\max a_y - b\pi_0(y)$ is non-empty, in which case the dynamics converges toward the uniform distribution on this set. This is notably the case when $\pi_0$ is uniform.

## A.2  Proof of Theorem 4.3

We now prove Theorem 4.3:

**Theorem 4.3.** *[Policy improvement dynamics]  Let $\mu$ be any policy with support $Y$, and let $V < V^\mu$.*

1. *Each application of the AsymRE algorithm increases the expected reward: $V^{\mathcal{T}_V\mu} \geq V^\mu$.*

2. *The sequence of expected rewards $V^{(\mathcal{T}_V)^n\mu}$ converges to some limit expected reward $V^\infty$. Let $Y^\infty \overset{def}{=} \{y \; : \; r(y) = V^\infty\}$. Then the mass of $(\mathcal{T}_V)^n\mu$ concentrates exponentially fast on $Y^\infty$, i.e.*

$$\sum_{y \notin Y^\infty} ((\mathcal{T}_V)^n\mu)(y) \leq c^n$$

*for some $c < 1$.*

3. *There exists $V_{0,\mu}$ such that the corresponding limit reward is optimal (i.e. $V^\infty = \max_{y \in Y} r(y)$) if and only if $V < V_{0,\mu}$.*

*Proof.* Let us first prove that the value increases. Let us denote $\pi = \mathcal{T}_V\mu$, $Y_*$ its support, and $\tau = \tau_{\mu,V}$ (defined as in Theorem 4.2). We have

$$(V^\mu - V)\pi(y) = (\mu(y)(r(y) - V) - \tau)^+.$$

This implies that for any $y \in Y_*$,

$$r(y) = \frac{(V^\mu - V)\pi(y) + \tau}{\mu(y)} + V.$$

The value of the policy $\pi$ can be computed as

$$V^\pi = \sum \pi(y)r(y) = \sum \pi(y)\left(\frac{(V^\mu - V)\pi(y) + \tau}{\mu(y)} + V\right)$$

$$= V + (V^\mu - V)\sum \frac{\pi(y)^2}{\mu(y)} + \tau \sum \frac{\pi(y)}{\mu(y)}$$

Using Cauchy-Schwarz, we have

$$\sum \frac{\pi(y)^2}{\mu(y)} = \sum \frac{\pi(y)^2}{\mu(y)} \sum \mu(z) \geq \left(\sum \frac{\pi(y)}{\sqrt{\mu(y)}}\sqrt{\mu(y)}\right)^2 = 1.$$

As a consequence, we deduce

$$V^\pi \geq V^\mu + \tau \sum \frac{\pi(y)}{\mu(y)} \geq V^\mu.$$

Where we have used the fact that $\tau \geq 0$.

After one application of $\mathcal{T}_V$, we only keep in the support of $\mathcal{T}_V\mu$ the arms for which $r(y) > V + \tau/\mu(y) \geq V$ (and they do not reappear in the support of $\mathcal{T}_V^n\mu$ for $n \geq 1$). This implies that $\tau_{\mathcal{T}_V^n\mu,V} = 0$ for $n \geq 1$ according to the characterization of $\tau$ provided in the proof of Theorem 4.2. This, in turn, implies

$$\frac{\mathcal{T}_V\mu(y)}{\mathcal{T}_V\mu(z)} = \frac{\mu(y)}{\mu(z)}\frac{r(y) - V}{r(z) - V}$$

for all $y, z$ in the support of $\mathcal{T}_V\mu$. Let denote

$$Y^\infty = \arg\max_{y \in \text{supp } \mathcal{T}_V\mu} r(y)$$

By recursion, we deduce, for $y \notin Y^\infty$ and $z \in Y^\infty$ both in the support of $\mathcal{T}_V \mu$, that

$$\frac{\mathcal{T}_V^n \mu(y)}{\mathcal{T}_V^n \mu(z)} = \frac{\mu(y)}{\mu(z)} \left( \frac{r(y) - V}{r(z) - V} \right)^n,$$

which implies that $\mathcal{T}_V^n \mu(y)$ goes exponentially fast to $0$ as $n$ increases.

The last statement of the theorem is a direct consequence of the characterization of $Y^\infty$, the monotonicity of the support of $\mathcal{T}_V \mu$ as $V$ increases, and the fact that for $V < \min r(y)$, $\mathrm{supp}(\mathcal{T}_V \mu) = Y$. $\qquad \square$

## B  Details regarding our LLMs experiments

We provide additional details regarding our experimental setup from Subsection 5.2.

We trained Llama-3.1-8B-Instruct with the AdamW optimizer Loshchilov and Hutter [2017] with a learning rate of $6 \times 10^{-8}$. Each time a prompt is sampled from the dataset, 8 trajectories are generated from the behavior policy and used to compute the empirical batch average $\overline{V}$. 128 trajectories are included in each gradient step. The maximum trajectories length is set to 2048 tokens (the generation is stopped after this number of tokens is reached).

The GPUs were split into two categories: workers that generated the responses to the prompt sampled from the dataset, and trainers that fine-tuned the model on the generated trajectories. In order to control the degree of off-policyness of the model, we varied the *update interval* parameter, which is the number of gradient steps between two updates of the weights of the workers. The larger this parameter, the more off-policy the training is.

Inference parameters were set to temperature 1.0 (respectively 0.1) and top-p 1.0 (respectively 0.95) for the training (respectively for the evaluation).

## C  Additional experiments

We report additional experiments.

**Entropies of the policies in the bandits setting**  In complement to Figure 1b, we report in Figure 7 the evolution of the entropy of the policy $\pi_t$ under the AsymRE algorithm in our bandits experiment.

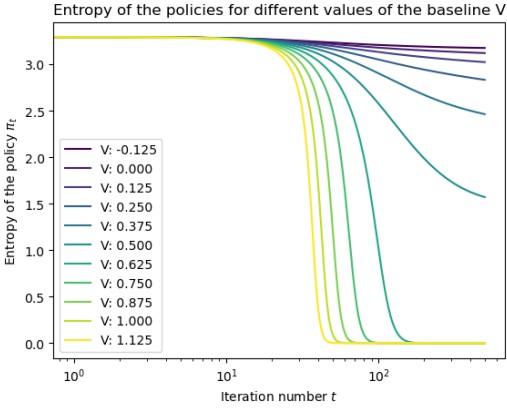

Figure 7: Entropy of the policy $\pi_t$ as a function of the iteration number $t$, for different values of the baseline parameter $\delta V$. For the larger values of $\delta V$, the entropy of the policies drops close to $0$ as the policy becomes deterministic.

**Entropies of the policies in the LLMs setting**  We report in Figure 8 the evolution of the entropy of the policy $\pi_t$ under the AsymRE algorithm in the LLMs setting.

**Additional experiments with Llama 3B**  In Figure 9, we represent the training dynamics (as in Figure 3) for Llama 3B when training on the MATH dataset.

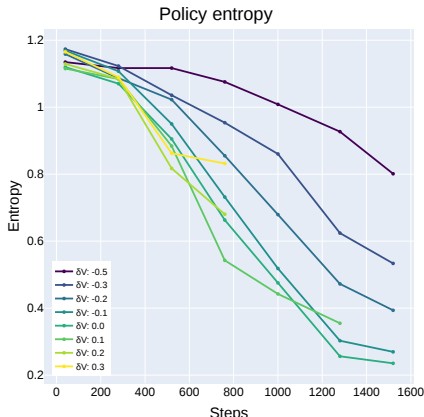

Figure 8: Evolution of the policy entropy during the training of Llama 8B on the MATH dataset (averaged over 3 seeds). The larger the baseline $V$ is, the faster the entropy decreases.

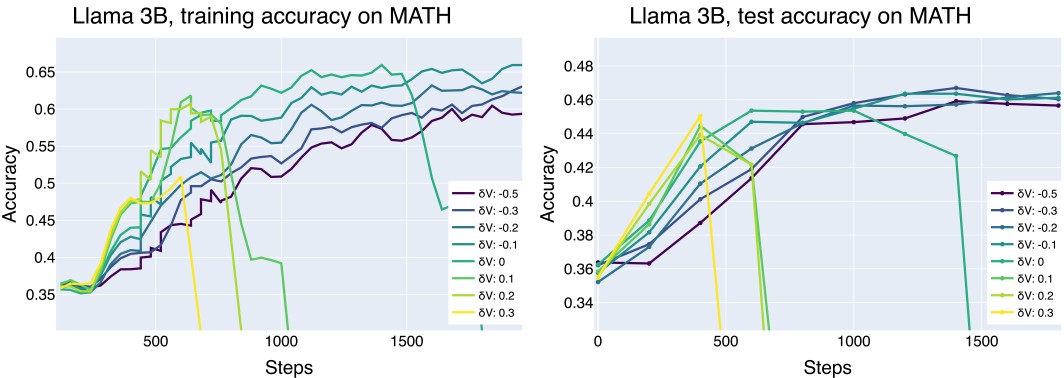

Figure 9: Training dynamics of Llama 3B on the MATH dataset (a moving average with a window of size 3 is applied to the training curve). The behavior policy is updated every $N = 250$ training steps.

**Additional experiments with NuminaMath**  In Figure 10, Figure 11 and Figure 12, we represent the training dynamics for Llama 8B, Qwen 3B and Llama 3B respectively when training and evaluating on subsets of size 142k and 2k respectively of the NuminaMath dataset (rather than on the MATH dataset).

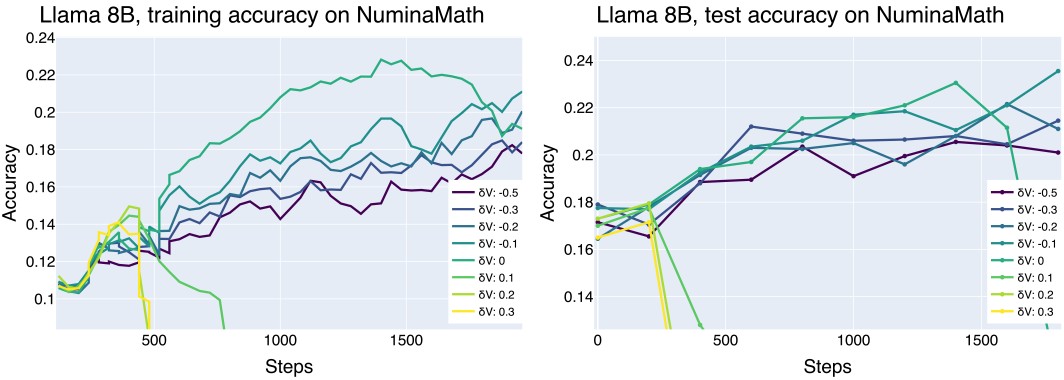

Figure 10: Training dynamics of Llama 8B on the NuminaMath sub-dataset (a moving average with a window of size 3 is applied to the training curve). The behavior policy is updated every $N = 250$ training steps.

**Additional comparisons to GRPO**  We report in Figure 13 a comparison between GRPO and AsymRE where the models are trained and tested on subsets of the NuminaMath dataset.

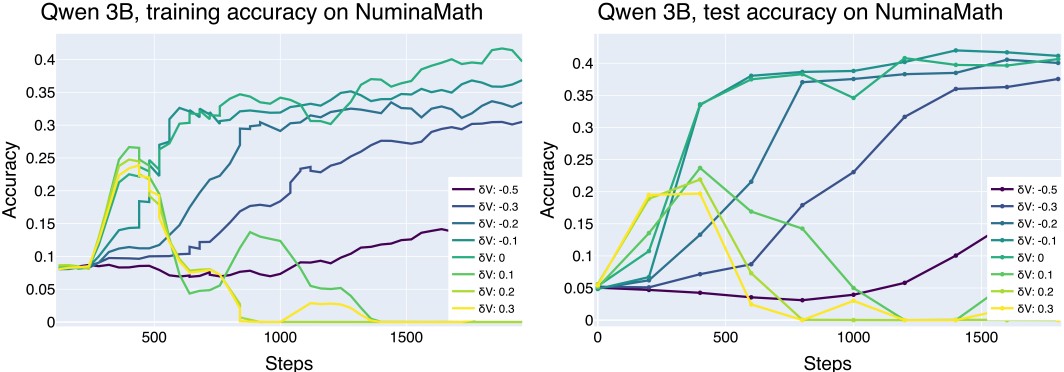

Figure 11: Training dynamics of Qwen 3B on the NuminaMath sub-dataset (a moving average with a window of size 3 is applied to the training curve). The behavior policy is updated every $N = 250$ training steps.

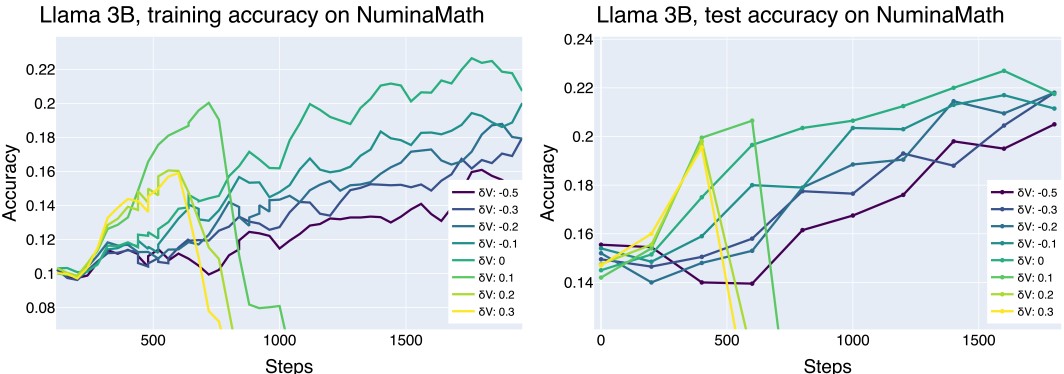

Figure 12: Training dynamics of Llama 3B on the NuminaMath sub-dataset (a moving average with a window of size 3 is applied to the training curve). The behavior policy is updated every $N = 250$ training steps.

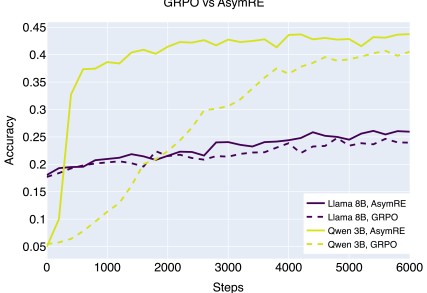

Figure 13: Test accuracy of Llama 8B and Qwen 3B trained on a subset of the NuminaMath dataset with GRPO and AsymRE (with $V = -0.1$). The behavior policy is updated every $N = 250$ steps, and a moving average with a window of size 3 is applied to the training curve.

