# OpenReview forum: "Asymmetric REINFORCE for off-Policy Reinforcement Learning: Balancing positive and negative rewards"
_NeurIPS.cc/2025/Conference — NeurIPS 2025 poster_

### Official Review · Reviewer_oGYm · 2025-06-30

**Clarity:** 3
**Significance:** 3
**Originality:** 3
**Rating:** 5
**Confidence:** 4

**Summary:**

The paper 'Reinforcement Learning From Others' Successes, Not Their Failures' propose to study the off-policy RL surrogate objective and discuss the implication of different choice of baseline value function $V$. The paper develops a clean theoretical analysis of the off-policy vanilla policy gradient (off-VPG) metho. Also provide a simple off-policy algorithm for LLM fine-tuning with almost no engineering overhead. Experimental validation verifies the finding that a small baseline $V$ benefits the training.

**Questions:**

1. First, I'm curious why the authors chose to study the surrogate loss function (1). As the authors claimed "As it does not involve any off-policy correction, such as importance sampling, we do not expect this algorithm to converge to an optimal policy in general." If the purpose is to study the off-policy algorithm as close to what it is in practice, I'm wondering if the analysis will be altered if a KL regularized or clipped objective is used (as in PPO objective).
2. It seems not very clear to me what Theorem 4.2 implies. In particular, when $V<V^{\mu}$, the optimal policy seems to be a reweighting of the sample policy $\mu$ by reward minus value $r(y)-V$, but for $V\geq V^{\mu}$ I feel it very hard to interpret the results.
3. What if we do adam or other adaptive algorithm in the experiments? Would the effect of different $V$ be mitigated?
4. The "Intuition" paragraph is indeed intuitive, however for LLM finetuning, we often have a continuous reward value than a binary reward. Could the authors comment on the implication of the theory for this kind of rewards?
5. Does the same collapse phenomenon observed in on-policy algorithms? I'm also curious on how the proposed method performs comparing to on-policy algorithms?

**Ethical Concerns:**

["NO or VERY MINOR ethics concerns only"]

**Limitations:**

The limitations are reflected in the questions section.

I don't think there are any potential negative societal impact for this work.

**Quality:**

3

**Strengths And Weaknesses:**

Strengths: The paper has clear motivation (off-policy RL) and theoretical analysis. The experiments (both bandit and LLM) are clear and verify the theoretical findings with clear evidence.

Weakness (please respond to the Questions section directly): Unclear why only analyzing the surrogate loss; The theoretical analysis requires more intepretation as it's not intuitive for now; The experiments might not be adequate and requires further comparison with on-policy algorithms.

---

> ### Author Rebuttal · Authors · 2025-07-31
>
> We are very thankful to the reviewers for their time and for their insightful comments.
>
> While the reviewers seem to agree that the papers' contributions are impactful, two shared concerns appear:
> * a lack of diversity in the experimental setup (not enough models, not enough datasets), and
> * a lack of comparison to popular RL algorithms.
>
> We want to address these shortcomings in this general response (which we include in all individual responses), before answering each reviewer's questions separately.
>
> Our initial intent was for the article to be equally balanced between a theoretical section, a bandits section, and a section on real-world applications, which is why we chose to keep the experimental section concise. However, the reviewers' comments have helped us realize that the paper would be strengthened by the inclusion of additional experiments, and we thank them for that.
>
> We have run a series of new experiments, which we hope will fully answer their concerns. In particular, we add two additional models (Qwen 2.5 3B and Llama 3.2 3B), a much larger dataset (NuminaMath) and a comparison to GRPO. The resulting tables can be found below (sadly, a recent change in the conference's rules does not allow us to share graphs anymore):
>
> **Table 1: Training Dynamics of the off-VPG Algorithm Across Different Baselines.**
> This extends our Figure 3 with additional models and datasets (Same hyperparameters. Reported values are the maximum pass@1 accuracy (in %) on the test set observed during training (2000 steps, with evaluation every 200 steps). Some runs crash during training, as in Figure 3; these are marked in bold.)
>
> |Dataset|Model|δV=-0.5|δV=-0.3|δV=-0.2|δV=-0.1|δV=0|δV=0.1|δV=0.2|δV=0.3|
> |-------|-----|--------|--------|--------|--------|-----|-----|-----|-----|
> |MATH|Llama8B|48.7|49.1|49.3|49.3|**48.9**|**47.5**|**46.1**|**43.5**|
> ||Llama3B|45.9|46.7|46.4|46.4|**45.4**|**44.5**|**43.9**|**45.0**|
> ||Qwen3B|62.2|63.5|65.1|65.6|64.9|**37.8**|**14.7**|**13.0**|
> |NuminaMath|Llama8B|20.5|21.5|22.2|23.6|**23.1**|**17.8**|**18.0**|**17.2**|
> ||Llama3B|20.5|21.8|21.8|21.7|22.7|**20.7**|**19.8**|**19.6**|
> ||Qwen3B|20.3|37.6|40.6|42.0|40.8|**23.7**|**21.9**|**19.7**|
>
> In our article, Figure 3 provides empirical evidence of how the choice of baseline δV affects the performance of our off-VPG algorithm.
> We extend the same experiment to two additional models (Llama3B and Qwen3B) and an additional dataset (a 144k-token subset of NuminaMath)
>
> We consistently observe the same two phenomena highlighted in Figure 3:
>
> - When the baseline is large (δV ≳ 0), training becomes unstable and often crashes (bold entries in **Table 1**).
> - As the baseline increases, the training becomes faster—reflected in the average left-to-right progression of higher scores within each row (for runs that did not crash).
>
> The general appearance of the training and test accuracies are also coherent with our earlier experiments. These results reinforce the robustness of our recommendation: to use a slightly negative baseline, such as δV = -0.1.
>
> **Table 2: Comparison of off-VPG and GRPO, off-policy**
> (10000 steps, with evaluation every 200 steps)
>
> |Dataset (Train & Test)|Model|off-VPG (off-policy)|GRPO (off-policy)|
> |------------------|------|--------------------------|----------------|
> |MATH|Llama8B|52.0|50.9|
> ||Llama3B|49.0|49.1|
> ||Qwen3B|64.8|61.7|
> |NuminaMath|Llama8B|26.1|24.8|
> ||Llama3B|25.5|25.1|
> ||Qwen3B|44.3|40.8|
>
> We conduct in **Table 2** a performance comparison between GRPO and off-VPG.
> We use the GRPO variant from (DeepSeekMath: Pushing the Limits of Mathematical Reasoning in Open Language Models, Shao et al), with KL regularization coefficient β = 0.001 and symmetric clipping at 0.2.
> For off-VPG, in line with our previous conclusions, we use a fixed baseline of δV = -0.1.
>
> Under these conditions, off-VPG appears to perform slightly better on average than GRPO, despite being more simple. All training runs are stable. We also observe that off-VPG achieves faster score improvements than GRPO, especially when training Qwen3B.
>
> Note that we do not yet claim that off-VPG is universally superior to GRPO--a fully rigorous comparison is beyond the scope of this work. Rather, we argue that its performance is competitive and reliable across diverse setups (three models and two datasets), and we plan on pursuing a more systematic investigation of this comparison in future work.
>
> If the article is accepted, we plan on including two additional graphs in the main text (a variant of Figure 3 with Qwen3B, and a comparison between off-VPG and GRPO with Llama 8B), and to put the other results in the Appendix, as they do not change the article's overall narrative (though they make our conclusions more robust).
>
> Note that we are very willing to run additional experiments in the remaining discussion period, should the reviewers deem it necessary.
>
> --- End of general response ---
>
>
> * First, I'm curious why the authors chose to study the surrogate loss function (1). [...] I'm wondering if the analysis will be altered if a KL regularized or clipped objective is used (as in PPO objective).
>
> *It was precisely the simplicity of the proposed loss which raised our interest: we wanted to study the potential impact of the simplest lever of action that we could think of, namely our baseline V. Adding a KL regularization or an importance ratio would certainly dramatically impact the theoretical analysis, though we expect that a KL regularization would mainly have the effect of keeping the trained policy closer to th reference policy, as in other algorithms. We leave this for future work. Note however that we have included preliminary comparisons to GRPO in our general response (Tables 2 and 3), where our loss seems to be competitive.*
>
> * It seems not very clear to me what Theorem 4.2 implies. In particular, when $V<V^\mu$, the optimal policy seems to be a reweighting of the sample policy $\mu$ by reward minus value $r(y)-V$, but for $V\geq V^\mu$ I feel it very hard to interpret the results.
>
> *In the case $V<V^\mu$, the limit policy (which is not necessarily optimal) is more precisely the positive part of a reweighting of $\mu$ minus a constant. The key takeaway is that the probability mass of an action $y$ w.r.t. the limit policy is a somewhat complicated increasing function of $\mu(y)(r(y)-V)$.
> When $V\geq V^\mu$, the limit policy is generically going to be a singleton (generically w.r.t. the choice of rewards and of $\mu$): large $V$s lead to a loss in diversity in the limit policy.*
>
> * What if we do adam or other adaptive algorithm in the experiments? Would the effect of different $V$ be mitigated?
>
> *In fact, our LLMs experiments are already conducted using AdamW, as is common when training large transformers (this is stated in Appendix B). Figure 4 shows that the effect is not mitigated.*
>
> * The "Intuition" paragraph is indeed intuitive, [...] for this kind of rewards?
>
> *All of our theorems hold for continuous rewards (we never assume that the rewards are discrete). We only give the example of $\{0,1\}$ rewards to help the readers' intuition, but the general idea is the same in the case of continuous rewards: letting $V$ be large puts comparatively more emphasis on the trajectories with the lowest rewards, and vice versa.*
>
>
> * Does the same collapse phenomenon observed in on-policy algorithms? I'm also curious on how the proposed method performs comparing to on-policy algorithms?
>
> *Regarding the impact of the baseline $\delta V$ on the risk of collapse, we have run some very preliminary experiments in an on-policy setting and observed similar results, but the risk of collapse seems to be more dependent on the choice of model, hyperparameters, and even possibly codebases (as many efficient RL codebases are never 100% on-policy to minimize the downtime of the worker Gpus tasked with generating training trajectories). These are still preliminary experiments from which we do not draw definitive conclusions.
> We have also run a comparison with GRPO in an on-policy setting, reported in ***Table 3*** below.*
>
>
> **Table 3: Comparison between off-VPG and GRPO, on-policy**
> (10000 steps with evaluation every 200 steps, bold values indicate that the run collapsed during training)
>
> |Model|off-VPG seed 1|seed 2|GRPO seed 1|seed 2|
> |--------------|----------------|--------|--------------|--------|
> |Qwen2.5-3B|65.4|64.2|**61.8**|**61.9**|
> |Llama3.1-8B|52.0|52.4|50.0|**50.1**|
>
> Though it it not the main focus of our work, we also conducted a comparison between the off-VPG algorithm and GRPO in an on-policy settings to assess whether off-VPG remains competitive. The results are presented in **Table 3**. The experiment uses the MATH dataset, with hyperparameters identical to those used in **Table 2**, except for the update frequency of the actor policy. In this experiment, updates occur at every gradient step, unlike previous experiments where updates occurred every 250 steps.
>
> We run two seeds for each configuration and report the highest score achieved on the test set. All learning curves increase steadily and plateau after approximately 3000 steps. The plateau is consistently higher for off-VPG than for GRPO, across both seeds and models. We also observe a sharp performance collapse during training in 3 out of 4 GRPO runs (highlighted in bold in the table), while no such collapse occurs with off-VPG.
>
> These results suggest that off-VPG remains a robust and effective choice in on-policy settings, with the added benefit of a simple implementation.

---

> > ### Comment · Reviewer_oGYm · 2025-08-08
> >
> > I thank the authors for their detailed response. I'll maintain my positive score.

---

### Official Review · Reviewer_sBrs · 2025-07-02

**Clarity:** 3
**Significance:** 3
**Originality:** 3
**Rating:** 4
**Confidence:** 3

**Summary:**

The paper analyses a **simple off-policy REINFORCE algorithm (off-VPG)** whose advantage is $A=r-V$, with a **user-tunable baseline V**.

- **Theory** For tabular soft-max policies the authors prove exponential convergence of off-VPG to a limit distribution $\pi^{*}_{\mu,V}$ (Theorem 4.2) and show a **phase-transition** at the critical value $V=V^{\mu}$ (the behavior-policy mean reward) whereby the policy’s support suddenly collapses to a singleton . Repeated application forms a policy-improvement scheme with guaranteed reward increase when $V<V^{\mu}$ (Theorem 4.3) .
- **Intuition** Lower $V$ accentuates high-reward (“success”) trajectories, higher $V$ emphasises pushing down low-reward (“failure”) ones; unlike the on-policy case, this baseline **changes the expected gradient direction off-policy** .
- **Experiments** (i) 100-arm bandit shows the predicted phase transition in reward, support, and entropy when $V$ crosses $V^{\mu}$ . (ii) Fine-tuning **Llama-3 8B** on the MATH dataset confirms that setting $\delta V \approx -0.1$ (i.e. slightly below the prompt-wise mean reward) yields stable learning, whereas $\delta V\ge 0$ causes catastrophic collapse .

The work offers a **computationally cheap knob**—just shift the baseline—to trade off stability versus exploration when using off-policy trajectories for LLM RLHF.

**Questions:**

1. Lines 596–599: The Lyapunov function is called “$\Phi_t$ is bounded **above** on the simplex”. Strictly speaking $F$ is concave when $b>0$; here “above” could be re-worded as “has a finite supremum”.
2. Lines 628–631: The bound $\|\nabla^2F\|\le2b$ → step-size $<1/b$. Readers may ask why the factor 2 disappears; worth stating that *any* constant $< 2/b$ suffices, but $1/b$ is chosen for margin.
3. In the third statement of Theorem 4.3, does the author mean that by setting the value of $V$ within a certain range, we can ensure that the policy value converges to the optimal value? If so, the use of the phrase “if and only if” is quite confusing.

**Ethical Concerns:**

["NO or VERY MINOR ethics concerns only"]

**Final Justification:**

The authors have addressed my concerns during the rebuttal phase. I maintain my positive evaluation.

**Limitations:**

1. Proofs rely on a finite-arm bandit with a tabular softmax policy; behavior with function approximation and delayed updates remains open.
2. The monotonic-improvement claim hinges on picking a baseline $V$ that is less than the unknown expected return $V^{\mu}$, yet in real-world RLHF that return is buried under reward noise, so telling practitioners to “just choose a slightly smaller $V$” simply dumps the toughest task back onto them.

**Quality:**

3

**Strengths And Weaknesses:**

**Strengths**

1. **Rigorous characterization** of off-VPG dynamics; explicit closed-form of $\pi^{*}_{\mu,V}$ and monotone support shrinkage clarify how baseline controls bias/variance trade-off .
2. **Actionable guideline**: choose $V$ slightly below the behavior—policy mean to avoid support collapse—validated on both bandits and an 8B LLM .
3. **Engineering simplicity**: no importance sampling, no critic, minimal code changes, yet competitive stability.
4. Clear discussion of why on-policy intuition fails off-policy and how this links to practical RLHF sampling delays .

**Weaknesses**

1. **Limited model and task diversity**: only one LLM architecture (Llama-3 8B) and one dataset (MATH).
2. **Missing baselines**: lacks direct comparison to IS-PPO, GRPO, or regularized PG, so relative performance ceiling is unclear.
3. **Statistical under-reporting**: LLM results use three seeds without CIs; bandit plots omit error bars.
4. **Code not yet released**, reducing reproducibility.

---

> ### Author Rebuttal · Authors · 2025-07-31
>
> We are very thankful to the reviewers for their time and for their insightful comments.
>
> While the reviewers seem to agree that the papers' contributions are impactful, two shared concerns appear:
> * a lack of diversity in the experimental setup (not enough models, not enough datasets), and
> * a lack of comparison to popular RL algorithms.
>
> We want to address these shortcomings in this general response (which we include in all individual responses), before answering each reviewer's questions separately.
>
> Our initial intent was for the article to be equally balanced between a theoretical section, a bandits section, and a section on real-world applications, which is why we chose to keep the experimental section concise. However, the reviewers' comments have helped us realize that the paper would be strengthened by the inclusion of additional experiments, and we thank them for that.
>
> We have run a series of new experiments, which we hope will fully answer their concerns. In particular, we add two additional models (Qwen 2.5 3B and Llama 3.2 3B), a much larger dataset (NuminaMath) and a comparison to GRPO. The resulting tables can be found below (sadly, a recent change in the conference's rules does not allow us to share graphs anymore):
>
> **Table 1: Training Dynamics of the off-VPG Algorithm Across Different Baselines.**
> This extends our Figure 3 with additional models and datasets (Same hyperparameters. Reported values are the maximum pass@1 accuracy (in %) on the test set observed during training (2000 steps, with evaluation every 200 steps). Some runs crash during training, as in Figure 3; these are marked in bold.)
>
> |Dataset|Model|δV=-0.5|δV=-0.3|δV=-0.2|δV=-0.1|δV=0|δV=0.1|δV=0.2|δV=0.3|
> |-------|-----|--------|--------|--------|--------|-----|-----|-----|-----|
> |MATH|Llama8B|48.7|49.1|49.3|49.3|**48.9**|**47.5**|**46.1**|**43.5**|
> ||Llama3B|45.9|46.7|46.4|46.4|**45.4**|**44.5**|**43.9**|**45.0**|
> ||Qwen3B|62.2|63.5|65.1|65.6|64.9|**37.8**|**14.7**|**13.0**|
> |NuminaMath|Llama8B|20.5|21.5|22.2|23.6|**23.1**|**17.8**|**18.0**|**17.2**|
> ||Llama3B|20.5|21.8|21.8|21.7|22.7|**20.7**|**19.8**|**19.6**|
> ||Qwen3B|20.3|37.6|40.6|42.0|40.8|**23.7**|**21.9**|**19.7**|
>
> In our article, Figure 3 provides empirical evidence of how the choice of baseline δV affects the performance of our off-VPG algorithm.
> We extend the same experiment to two additional models (Llama3B and Qwen3B) and an additional dataset (a 144k-token subset of NuminaMath)
>
> We consistently observe the same two phenomena highlighted in Figure 3:
>
> - When the baseline is large (δV ≳ 0), training becomes unstable and often crashes (bold entries in **Table 1**).
> - As the baseline increases, the training becomes faster—reflected in the average left-to-right progression of higher scores within each row (for runs that did not crash).
>
> The general appearance of the training and test accuracies are also coherent with our earlier experiments. These results reinforce the robustness of our recommendation: to use a slightly negative baseline, such as δV = -0.1.
>
> **Table 2: Comparison of off-VPG and GRPO, off-policy**
> (10000 steps, with evaluation every 200 steps)
>
> |Dataset (Train & Test)|Model|off-VPG (off-policy)|GRPO (off-policy)|
> |------------------|------|--------------------------|----------------|
> |MATH|Llama8B|52.0|50.9|
> ||Llama3B|49.0|49.1|
> ||Qwen3B|64.8|61.7|
> |NuminaMath|Llama8B|26.1|24.8|
> ||Llama3B|25.5|25.1|
> ||Qwen3B|44.3|40.8|
>
> We conduct in **Table 2** a performance comparison between GRPO and off-VPG.
> We use the GRPO variant from (DeepSeekMath: Pushing the Limits of Mathematical Reasoning in Open Language Models, Shao et al), with KL regularization coefficient β = 0.001 and symmetric clipping at 0.2.
> For off-VPG, in line with our previous conclusions, we use a fixed baseline of δV = -0.1.
>
> Under these conditions, off-VPG appears to perform slightly better on average than GRPO, despite being more simple. All training runs are stable. We also observe that off-VPG achieves faster score improvements than GRPO, especially when training Qwen3B.
>
> Note that we do not yet claim that off-VPG is universally superior to GRPO--a fully rigorous comparison is beyond the scope of this work. Rather, we argue that its performance is competitive and reliable across diverse setups (three models and two datasets), and we plan on pursuing a more systematic investigation of this comparison in future work.
>
> If the article is accepted, we plan on including two additional graphs in the main text (a variant of Figure 3 with Qwen3B, and a comparison between off-VPG and GRPO with Llama 8B), and to put the other results in the Appendix, as they do not change the article's overall narrative (though they make our conclusions more robust).
>
> Note that we are very willing to run additional experiments in the remaining discussion period, should the reviewers deem it necessary.
>
> --- End of general response ---
>
> * Limited model and task diversity: only one LLM architecture (Llama-3 8B) and one dataset (MATH).
> and
> * Missing baselines: lacks direct comparison to IS-PPO, GRPO, or regularized PG, so relative performance ceiling is unclear.
>
> *We have ran additional experiments with 2 other models, one additional dataset, and a baseline (GRPO) (see the general response).*
>
> * Statistical under-reporting: LLM results use three seeds without CIs; bandit plots omit error bars.
>
> *Figure 3 was only ran with 3 seeds because training $8\cdot 3$ 8B models is quite expensive. Do you think that showing intervals representing the variance of these trajectories would improve the figure?
> In Figure 4, we show all trajectories for 7 different seeds to prove the robustness of the phenomenon studied to randomness.
> In the bandit setting, we study the expected version of the algorithm: as such, the gradient descent is non-stochastic. The only randomness comes from the drawing of the rewards associated to each arms, i.e. when definining the bandit task. Combining results corresponding to different tasks (e.g. with error bars) yields a graph that is very hard to interpret. We have observed that the general appearance of the curves is very stable w.r.t. the random seed; should we add variants of these curves for other random seeds in the Appendix?*
>
> * Code not yet released, reducing reproducibility.
>
> *Our code is shared with another team within our lab; we should be able to release it a month (perhaps even in time for the camera-ready version).*
>
> * Lines 596–599: The Lyapunov function is called “$\Phi_t$ is bounded above on the simplex”. Strictly speaking $\Phi_t$ is concave when $b>0$ ; here “above” could be re-worded as “has a finite supremum”.
>
>
> Thank you for this wording suggestion.
>
>  * Lines 628–631: The bound $|\nabla^2 F|\leq 2b$ $\rightarrow$ step-size $<1/b$. Readers may ask why the factor 2 disappears; worth stating that any constant $<2b$ suffices, but $1/b$ is chosen for margin.
>
> Thank you for pointing out this potential confusion. The factor of 2 disappears because we are using the bound on the step size, $< 2 / \| \nabla^2 F \|$. We are not aware of a tighter result.
>
>
> * In the third statement of Theorem 4.3, does the author mean that by setting the value of $V$ within a certain range, we can ensure that the policy value converges to the optimal value? If so, the use of the phrase “if and only if” is quite confusing.
>
> *As you guessed, the statement should be understood as "the limit policy is optimal if and only if $V<V_{0,\mu}$". We agree that the phrasing is awkward, and will change it.*
>
>
> * Proofs rely on a finite-arm bandit with a tabular softmax policy; behavior with function approximation and delayed updates remains open.
> *The behaviour with respect to delayed updates is described in the case of finite-arm bandits by Thm 4.3. We agree that going from a tabular setting to the case of LLMs is a non-trivial change, which our theory does not fully capture (though our experiments show that the intuition from bandits still applies in the case of LLMs). In our defense, there is no agreed upon mathematical framework that convincingly captures the regularity of transformers with respect to the prompts $x$ (e.g. the fact that similar prompts are treated similarly), and such a framework would be needed to reach theoretical conclusions regarding LLMs.*
>
> * The monotonic-improvement claim hinges on picking a baseline $V$ that is less than the unknown expected return $V^\mu$, yet in real-world RLHF that return is buried under reward noise, so telling practitioners to “just choose a slightly smaller $V$” simply dumps the toughest task back onto them.
>
> *In the case of LLMs, which is our main motivation, this problem is partially averted thanks to the additive renormalization by the prompt-dependent average reward $V^{\mu(\cdot |x)}$, which we estimate with the group-average $\hat{V} = \frac{1}{G}\sum_{i=1}^G r(y_i,x)$:
> $\mathbb{E}[\frac{1}{G} \sum_{i=1}^{G} (r(y_i,x) - (\hat V + \delta V)) \log(\pi(y_{i}|x))]$, where the expectation is w.r.t. ${x \sim D, \{y_i\}_{i=1}^{G} \sim \mu(.|x)}$.
> As an estimator of $V^{\mu(\cdot |x)}$ is substracted to the rewards for prompt $x$, the expected return after the renormalization is $0$, which means that the critical value now becomes $\delta V = 0$ uniformly across the prompts $x$s.
> Our experiments (including the new ones) show that $\delta V =-0.1$ seems like a safe bet, though we do not claim that it is optimal.*

---

> > ### Comment · Reviewer_sBrs · 2025-08-04
> >
> > Thank you for your response. I don't have further questions.

---

### Official Review · Reviewer_ARzT · 2025-07-02

**Clarity:** 2
**Significance:** 3
**Originality:** 2
**Rating:** 3
**Confidence:** 3

**Summary:**

This work discusses the design of off-policy RL for LLMs finetuning. By introducing a baseline V, the paper discusses the impact of policy convergence and explores how different baseline settings affect model performance. The conclusion drawn is that higher optimization weights should be given to positive trajectories, achieved through setting a baseline slightly lower than the expected value of the behavior policy.

**Questions:**

1. There are relevant discussions on the influence of bias in off-policy contexts within Off-Policy Actor-Critic[1]. It would be better if a reasonable citation were added.
2.  Off-VPG does not account for the relationship between behavior policy and current policy when utilizing trajectories. Off-policy policy gradient algorithms often correct bias through the importance weights. Should this be considered as a baseline? For instance, naive Off-Policy PG [2].
3. Should the experiments be conducted on larger datasets to further validate the effectiveness of the Off-VPG? Additionally, the evaluation is only performed on the MATH dataset without assessing the LLM's performance in general textual tasks or other mathematical benchmarks. Lack a more comprehensive experimental validation.
4. Some specific implementation details for Off-VPG are missing. For example, is the behavior policy completely unupdated or updated with a delay? If it remains unupdated, could it degrade into rejection sampling fine-tuning?
5. There is an absence of comparisons between on- and off-policy algorithms concerning performance and sample efficiency to substantiate claims such as "Off-policy methods offer greater implementation simplicity and data efficiency than on-policy techniques."

[1]Off-Policy Actor-Critic. Thomas Degris, Martha White, and Richard S. Sutton. ICML 2012.

[2]Off-Policy PG: https://lilianweng.github.io/posts/2018-04-08-policy-gradient/

Trivial:
1. The organization and formatting of theorem sections make them difficult to read.
2. The x-axis title in Figure 2 lacks clarity.
3. Color differentiation in line graphs is too low. It takes effort to discern them.

**Ethical Concerns:**

["NO or VERY MINOR ethics concerns only"]

**Final Justification:**

This article offers a perspective on how to allocate good and bad data in off-policy data for the strength of LLM updates, which is worth studying. However, the author does not clearly delineate the prior work on off-policy RL in the article, lacks discussion on the applicability compared with on-policy methods in LLMs, and the experiments are insufficient to adequately illustrate the claim. Therefore, I still believe this paper is not ready for NeurIPS 2025 from both the presentation and experimental aspects.

**Limitations:**

yes

**Paper Formatting Concerns:**

No formatting issues.

**Quality:**

2

**Strengths And Weaknesses:**

Strengths:
Utilizing off-policy RL to optimize LLMs holds research value and may enhance the efficiency and deployment of RL-based fine-tuning. The theoretical analysis provides substantial support for the algorithm.

Weaknesses:
1. The conclusion that the baselines in off-policy learning not only influence the variance as in the on-policy algorithms but also introduce policy bias is not much novel in the off-policy RL scope. Some similar conclusions have been discussed in previous off-policy RL algorithms. Besides, the final conclusion (optimize more on the positive trajectories) quite aligns with the intuition. These factors somewhat diminish the contribution of the theoretical analysis *slightly*.
2. The off-VPG is evaluated merely with one model on one dataset. Thus, the experimental support for the algorithm is limited.
3. There is no comparison or discussion regarding on-policy finetuing. Although off-policy methods may have advantages in efficiency or implementation, is there a trade-off between performance and efficiency?
4. This off-policy algorithm design does not specifically target LLMs and lacks discussions related to existing off-policy RL algorithms.

---

> ### Author Rebuttal · Authors · 2025-07-31
>
> We are very thankful to the reviewers for their time and for their insightful comments.
>
> While the reviewers seem to agree that the papers' contributions are impactful, two shared concerns appear:
> * a lack of diversity in the experimental setup (not enough models, not enough datasets), and
> * a lack of comparison to popular RL algorithms.
>
> We want to address these shortcomings in this general response (which we include in all individual responses), before answering each reviewer's questions separately.
>
> Our initial intent was for the article to be equally balanced between a theoretical section, a bandits section, and a section on real-world applications, which is why we chose to keep the experimental section concise. However, the reviewers' comments have helped us realize that the paper would be strengthened by the inclusion of additional experiments, and we thank them for that.
>
> We have run a series of new experiments, which we hope will fully answer their concerns. In particular, we add two additional models (Qwen 2.5 3B and Llama 3.2 3B), a much larger dataset (NuminaMath) and a comparison to GRPO. The resulting tables can be found below (sadly, a recent change in the conference's rules does not allow us to share graphs anymore):
>
> **Table 1: Training Dynamics of the off-VPG Algorithm Across Different Baselines.**
> This extends our Figure 3 with additional models and datasets (Same hyperparameters. Reported values are the maximum pass@1 accuracy (in %) on the test set observed during training (2000 steps, with evaluation every 200 steps). Some runs crash during training, as in Figure 3; these are marked in bold.)
>
> |Dataset|Model|δV=-0.5|δV=-0.3|δV=-0.2|δV=-0.1|δV=0|δV=0.1|δV=0.2|δV=0.3|
> |-------|-----|--------|--------|--------|--------|-----|-----|-----|-----|
> |MATH|Llama8B|48.7|49.1|49.3|49.3|**48.9**|**47.5**|**46.1**|**43.5**|
> ||Llama3B|45.9|46.7|46.4|46.4|**45.4**|**44.5**|**43.9**|**45.0**|
> ||Qwen3B|62.2|63.5|65.1|65.6|64.9|**37.8**|**14.7**|**13.0**|
> |NuminaMath|Llama8B|20.5|21.5|22.2|23.6|**23.1**|**17.8**|**18.0**|**17.2**|
> ||Llama3B|20.5|21.8|21.8|21.7|22.7|**20.7**|**19.8**|**19.6**|
> ||Qwen3B|20.3|37.6|40.6|42.0|40.8|**23.7**|**21.9**|**19.7**|
>
> In our article, Figure 3 provides empirical evidence of how the choice of baseline δV affects the performance of our off-VPG algorithm.
> We extend the same experiment to two additional models (Llama3B and Qwen3B) and an additional dataset (a 144k-token subset of NuminaMath)
>
> We consistently observe the same two phenomena highlighted in Figure 3:
>
> - When the baseline is large (δV ≳ 0), training becomes unstable and often crashes (bold entries in **Table 1**).
> - As the baseline increases, the training becomes faster—reflected in the average left-to-right progression of higher scores within each row (for runs that did not crash).
>
> The general appearance of the training and test accuracies are also coherent with our earlier experiments. These results reinforce the robustness of our recommendation: to use a slightly negative baseline, such as δV = -0.1.
>
> **Table 2: Comparison of off-VPG and GRPO, off-policy**
> (10000 steps, with evaluation every 200 steps)
>
> |Dataset (Train & Test)|Model|off-VPG (off-policy)|GRPO (off-policy)|
> |------------------|------|--------------------------|----------------|
> |MATH|Llama8B|52.0|50.9|
> ||Llama3B|49.0|49.1|
> ||Qwen3B|64.8|61.7|
> |NuminaMath|Llama8B|26.1|24.8|
> ||Llama3B|25.5|25.1|
> ||Qwen3B|44.3|40.8|
>
> We conduct in **Table 2** a performance comparison between GRPO and off-VPG.
> We use the GRPO variant from (DeepSeekMath: Pushing the Limits of Mathematical Reasoning in Open Language Models, Shao et al), with KL regularization coefficient β = 0.001 and symmetric clipping at 0.2.
> For off-VPG, in line with our previous conclusions, we use a fixed baseline of δV = -0.1.
>
> Under these conditions, off-VPG appears to perform slightly better on average than GRPO, despite being more simple. All training runs are stable. We also observe that off-VPG achieves faster score improvements than GRPO, especially when training Qwen3B.
>
> Note that we do not yet claim that off-VPG is universally superior to GRPO--a fully rigorous comparison is beyond the scope of this work. Rather, we argue that its performance is competitive and reliable across diverse setups (three models and two datasets), and we plan on pursuing a more systematic investigation of this comparison in future work.
>
> If the article is accepted, we plan on including two additional graphs in the main text (a variant of Figure 3 with Qwen3B, and a comparison between off-VPG and GRPO with Llama 8B), and to put the other results in the Appendix, as they do not change the article's overall narrative (though they make our conclusions more robust).
>
> Note that we are very willing to run additional experiments in the remaining discussion period, should the reviewers deem it necessary.
>
> --- End of general response ---
>
>
> * The conclusion that the baselines [...] contribution of the theoretical analysis slightly.
>
> *It is true that the observation that baselines introduce biases is not novel; our main contribution here is a theoretical analysis of the effects of such a bias, as well as empirical confirmations that such biases can have a positive impact.
> Regarding the final conclusion, it is in fact very common to hear that negative trajectories can be just as important, or even more important than positive ones (see e.g. [1]). A common missconception is that penalizing negative samples has the effect of encouraging diversity and exploration, because the probability mass diverted from the penalized action would be somewhat uniformly distributed over other actions. This is in fact not the case, as shown by our results: when penalizing a negative action, most of the probability mass goes to the modes of the distribution $\pi$, as visible in the proof of Thm 4.2, which leads to the lack of diversity described in Thm 4.2 ("the rich get richer").
> [1] The Surprising Effectiveness of Negative Reinforcement in LLM Reasoning, Zhu et al.*
>
> * The off-VPG is evaluated merely with one model on one dataset. Thus, the experimental support for the algorithm is limited.
> * and
> * Should the experiments be [...] Lack a more comprehensive experimental validation.
>
> *We agree; see the additional experiments in our general response.*
>
> * This off-policy algorithm design does not specifically target LLMs and lacks discussions related to existing off-policy RL algorithms.
>
> *We see the simplicity of our algorithm (combined with its good performance), which is not specific to LLMs, as an advantage. Are there any particular works that you think would be good additions to our related works section (besides [1])?*
>
> * There are relevant discussions [...].
>
> *Thank you very much, we will be sure to add this work to our related works section.*
>
> * Off-VPG does not account for the relationship between behavior policy and current policy [...] For instance, naive Off-Policy PG [2].
>
> *It is common in off-policy RL to correct bias with an importance ratio as follows $\mathrm{E}_{y \sim \mu}[\frac{\pi(y)}{\mu(y)}A(y)]$, where $A(y)$ is some advantage term (e.g. the reward itself). In the context of LLMs, the action $y$ is a long sequence of tokens $t_1,\ldots, t_k$, and the importance ratio $\frac{\pi(y)}{\mu(y)} = \frac{\pi(t_1) \pi(t_2|t_1)\ldots}{\mu(t_1) \mu(t_2|t_1)\ldots}$ is often almost $0$ or extremely large when $\pi \neq \mu$, which makes it impractical. In practice, the ratio is usually clipped on a per-token basis, e.g. in PPO or GRPO. As GRPO is considered one of the sota losses, we compare it to our algorithm in our additional experiments (see the general response).*
>
>
> * Some specific implementation details for Off-VPG are missing. For example, [...] into rejection sampling fine-tuning?
>
> *In Definition 4.1, we define the algorithm that we call off-VPG, in which the behaviour policy remains fixed. The behaviour of this algorithm is described in Thm 4.2, and illustrated in the bandits setting in Figure 1.
> In the paragraph "Policy improvement scheme with off-VPG", we describe the derived policy improvement scheme in which the behaviour policy is updated with the (current) trained policy $\pi$ with a delay (i.e. every N gradient steps on $\pi$, we update $\mu \leftarrow \pi$), and which we also call "delayed updates setting". Thm 4.3 applies to this case, and Figures 2, 3 and 4 illustrate it in the bandits and the LLMs setting respectively.*
>
> * There is an absence of comparisons [...] than on-policy techniques."
> and
> * There is no comparison [...] between performance and efficiency?
>
> *Our phrasing might have been overly enthusiastic. The goal of this article was not to prove that off-policy RL is better than on-policy RL (e.g. in terms of performance/efficiency trade-off). Our point of view was rather: practitioners often want or have to train off-policy (to benefit from some of the advantages of off-policiness, or because they are constrained by their setup), which makes off-policy RL worth studying. We will rephrase this sentence in a more neutral way.
> Though a rigorous study of the trade-offs between on-policy and off-policy RL is beyond the scope of this article and left to future work, we note that our preliminary experiments suggest that off-policy training (with off-VPG or with GRPO) with an update frequency of 250 gradient steps for the actor policy leads to final scores that are similar to those obtained with on-policy training (e.g. compare Tables 1 and 2 from our general response with Table 3 in our response to reviewer oGYm)*

---

> ### Comment · Reviewer_ARzT · 2025-08-04
>
> Thanks for the response. While I will raise the score to borderline reject, my view remains slightly negative because the experiments lack adequate evaluation of off-policy RL methods and comparison with potential on-policy baselines, leaving the claims insufficiently supported.

---

### Official Review · Reviewer_x4kd · 2025-07-03

**Clarity:** 4
**Significance:** 3
**Originality:** 4
**Rating:** 5
**Confidence:** 4

**Summary:**

This paper analyze off policy policy gradient (in the bandit setting) and shows how different baselines can affect the learned policy. They show when the baseline is lower than the behavior value, the learned policy improves; the behavior value is a critical point, using a baseline beyond which can lead to collapsing in policy support. The algorithm is tested in a synthesized bandit problem and training a 8B Llama. The experimental results support the theory's insight.

**Questions:**

1. In the analyses, are the rewards implicitly assumed to be non-negative?

2. In Theorem 4.3, does the set Y^inf exist always? since the policy can be stochastic.

3. I do not understand the statement of point 3 in Theorem 4.3 from the current writing, and reading the proof doesn't help. Can you expand more on what you wish to convey?

4. Do you have a bound on how large V^inf is? Maybe this is related to the misunderstanding above.

5. How does the convergence rate and V^inf change with V in Theorem 4.3? I

**Ethical Concerns:**

["NO or VERY MINOR ethics concerns only"]

**Limitations:**

Yes

**Quality:**

4

**Strengths And Weaknesses:**

S1. To my knowledge, despite many off-policy PG papers, this paper is the first to analyze directly off-policy without importance weight or state occupancy correction.

S2 The theoretical analyses reveal an interesting landscape of off-policy PG's dynamics and is quite complete. It is also good to see the experimental results corroborate with the theoretical analyses.

S3. The paper is overall well-written and well motivated. The analyses here are timely given that PG algorithms are gaining interests again in the LLM era. .

W1. The current analyses assumes a bandit setting, which is different form the contextual setting used in practice. However, I think this limitation is minor given the contribution of the paper.

W2. Some minor writing clarity can be improved.

---

> ### Author Rebuttal · Authors · 2025-07-31
>
> We are very thankful to the reviewers for their time and for their insightful comments.
>
> While the reviewers seem to agree that the papers' contributions are impactful, two shared concerns appear:
> * a lack of diversity in the experimental setup (not enough models, not enough datasets), and
> * a lack of comparison to popular RL algorithms.
>
> We want to address these shortcomings in this general response (which we include in all individual responses), before answering each reviewer's questions separately.
>
> Our initial intent was for the article to be equally balanced between a theoretical section, a bandits section, and a section on real-world applications, which is why we chose to keep the experimental section concise. However, the reviewers' comments have helped us realize that the paper would be strengthened by the inclusion of additional experiments, and we thank them for that.
>
> We have run a series of new experiments, which we hope will fully answer their concerns. In particular, we add two additional models (Qwen 2.5 3B and Llama 3.2 3B), a much larger dataset (NuminaMath) and a comparison to GRPO. The resulting tables can be found below (sadly, a recent change in the conference's rules does not allow us to share graphs anymore):
>
> **Table 1: Training Dynamics of the off-VPG Algorithm Across Different Baselines.**
> This extends our Figure 3 with additional models and datasets (Same hyperparameters. Reported values are the maximum pass@1 accuracy (in %) on the test set observed during training (2000 steps, with evaluation every 200 steps). Some runs crash during training, as in Figure 3; these are marked in bold.)
>
> |Dataset|Model|δV=-0.5|δV=-0.3|δV=-0.2|δV=-0.1|δV=0|δV=0.1|δV=0.2|δV=0.3|
> |-------|-----|--------|--------|--------|--------|-----|-----|-----|-----|
> |MATH|Llama8B|48.7|49.1|49.3|49.3|**48.9**|**47.5**|**46.1**|**43.5**|
> ||Llama3B|45.9|46.7|46.4|46.4|**45.4**|**44.5**|**43.9**|**45.0**|
> ||Qwen3B|62.2|63.5|65.1|65.6|64.9|**37.8**|**14.7**|**13.0**|
> |NuminaMath|Llama8B|20.5|21.5|22.2|23.6|**23.1**|**17.8**|**18.0**|**17.2**|
> ||Llama3B|20.5|21.8|21.8|21.7|22.7|**20.7**|**19.8**|**19.6**|
> ||Qwen3B|20.3|37.6|40.6|42.0|40.8|**23.7**|**21.9**|**19.7**|
>
> In our article, Figure 3 provides empirical evidence of how the choice of baseline δV affects the performance of our off-VPG algorithm.
> We extend the same experiment to two additional models (Llama3B and Qwen3B) and an additional dataset (a 144k-token subset of NuminaMath)
>
> We consistently observe the same two phenomena highlighted in Figure 3:
>
> - When the baseline is large (δV ≳ 0), training becomes unstable and often crashes (bold entries in **Table 1**).
> - As the baseline increases, the training becomes faster—reflected in the average left-to-right progression of higher scores within each row (for runs that did not crash).
>
> The general appearance of the training and test accuracies are also coherent with our earlier experiments. These results reinforce the robustness of our recommendation: to use a slightly negative baseline, such as δV = -0.1.
>
> **Table 2: Comparison of off-VPG and GRPO, off-policy**
> (10000 steps, with evaluation every 200 steps)
>
> |Dataset (Train & Test)|Model|off-VPG (off-policy)|GRPO (off-policy)|
> |------------------|------|--------------------------|----------------|
> |MATH|Llama8B|52.0|50.9|
> ||Llama3B|49.0|49.1|
> ||Qwen3B|64.8|61.7|
> |NuminaMath|Llama8B|26.1|24.8|
> ||Llama3B|25.5|25.1|
> ||Qwen3B|44.3|40.8|
>
> We conduct in **Table 2** a performance comparison between GRPO and off-VPG.
> We use the GRPO variant from (DeepSeekMath: Pushing the Limits of Mathematical Reasoning in Open Language Models, Shao et al), with KL regularization coefficient β = 0.001 and symmetric clipping at 0.2.
> For off-VPG, in line with our previous conclusions, we use a fixed baseline of δV = -0.1.
>
> Under these conditions, off-VPG appears to perform slightly better on average than GRPO, despite being more simple. All training runs are stable. We also observe that off-VPG achieves faster score improvements than GRPO, especially when training Qwen3B.
>
> Note that we do not yet claim that off-VPG is universally superior to GRPO--a fully rigorous comparison is beyond the scope of this work. Rather, we argue that its performance is competitive and reliable across diverse setups (three models and two datasets), and we plan on pursuing a more systematic investigation of this comparison in future work.
>
> If the article is accepted, we plan on including two additional graphs in the main text (a variant of Figure 3 with Qwen3B, and a comparison between off-VPG and GRPO with Llama 8B), and to put the other results in the Appendix, as they do not change the article's overall narrative (though they make our conclusions more robust).
>
> Note that we are very willing to run additional experiments in the remaining discussion period, should the reviewers deem it necessary.
>
> --- End of general response ---
>
> * In the analyses, are the rewards implicitly assumed to be non-negative?
>
> No, the analyses are invariant to any transformation of the rewards of the form $\alpha \cdot r + \beta$ (and the same holds for the baseline $V$).
>
> * In Theorem 4.3, does the set Y^inf exist always? since the policy can be stochastic.
>
> Theorem 4.3 assumes access to the full population of samples (i.e., we know $\mu$) and the ability to perfectly optimize our surrogate objective (i.e., we can apply ${\cal T}$ exactly). This removes the stochasticity usually referred to as approximation and optimization error. In this setting, even if $\pi$ is stochastic, $Y^\inf$ always exists. We did not address approximation and optimization errors, as they can be handled with classical techniques, which would lengthen the proofs without adding novel insights.
>
> * I do not understand the statement of point 3 in Theorem 4.3 from the current writing, and reading the proof doesn't help. Can you expand more on what you wish to convey?
>
> The phrasing is indeed a bit ambiguous; thank you for pointing it out, we will rephrase it.
> The third point of Theorem 4.3 states that the limit reward is going to be optimal if and only if the baseline $V$ is smaller than some threshold $V_{0,\mu}$ (which depends on the interplay between $\mu$ and $\pi^*$).
>
> * Do you have a bound on how large V^inf is? Maybe this is related to the misunderstanding above.
>
> We have not sought an explicit bound for $V^\inf$. A natural lower bound is $V^\inf > \max_y \mu(y) r(y)$.
>
> * How does the convergence rate and V^inf change with V in Theorem 4.3?
>
> This is a very good question that highlights a trade-off: if $V$ is too large, there is a risk of eliminating good arms (indeed, $V^\inf$ is a decreasing function of $V$); conversely, if $V$ is too small, the constant $c$ becomes small (indeed, $c$ is an increasing function of $V$). In other words, the larger $V$ is, the faster the convergence, but the higher the risk of converging to a suboptimal policy.

---

### Official Review · Reviewer_Wxir · 2025-07-05

**Clarity:** 2
**Significance:** 2
**Originality:** 3
**Rating:** 3
**Confidence:** 3

**Summary:**

RL has been very successful in aligning LLMs for various downstream tasks. However, the most popular RL methods are on-policy and have several drawbacks. This paper proposes a new off-policy method to address the limitations of on-policy RL methods when aligning LLMs. The paper proves two theorems: one regarding the limits of the algorithm in relation to various baselines, and the other concerning policy improvement. Additionally, the paper conducts several experiments using LLaMA on mathematical tasks. The results of the experiment show that the various baselines, as shown in the proof, empirically affect the models' downstream performance.

**Questions:**

- How does this paper compare to [1], PPO [2] in terms of the importance ratio enabling off-policy updates, and GRPO?
- The original motivation of the paper was efficiency, but there are no comparisons to on-policy algorithms to demonstrate efficiency in any way.
- For equation (1), how are you avoiding the importance sampling issue?
- How do you choose the N to use for delayed updates? How sensitive is the proposed algorithm with respect to N?
- Are the bounds in the bandit RL setting or the multi-step RL setting?
- Could you provide more motivation for why you have the additional delta V term? The fact that the value of V varies for each prompt is expected, so I'm not sure why adding the correction term is necessary.
- How is delta V trained in practice because it seems to be different than V^\mu, which is computed using multiple rollouts.
- For most of the bounds in the appendix, there are no assumptions made regarding the ratio of \pi(y)/\mu(y), which could be bad if the ratio is not assumed to be bounded, right?

[1] Fine-Tuning Language Models with Advantage-Induced Policy Alignment by Banghua Zhu et al. 2023

[2] Batch size-invariance for policy optimization by Jacob Hilton et al 2022

**Ethical Concerns:**

["NO or VERY MINOR ethics concerns only"]

**Limitations:**

Yes

**Quality:**

3

**Strengths And Weaknesses:**

Strengths:

The ability to train off-policy for LLMs alignment is a very important problem. The paper's proposal is scalable and has the potential to impact training models in an asynchronous manner. The paper comes with guarantees that hold in practice, which makes it a promising direction.

Weaknesses:

The paper has several shortcomings. First, it omits some key baseline algorithms necessary for a thorough comparison. The evaluation is also limited to a single model and task, which raises concerns about its robustness and applicability to other tasks. Moreover, without including baseline algorithms, it’s unclear how the proposed method compares to existing approaches in the literature, such as [1]. Finally, some of the theorems in the appendix seem underexplained.

[1] Fine-Tuning Language Models with Advantage-Induced Policy Alignment by Banghua Zhu et al. 2023

---

> ### Author Rebuttal · Authors · 2025-07-31
>
> We are very thankful to the reviewers for their time and for their insightful comments.
>
> While the reviewers seem to agree that the papers' contributions are impactful, two shared concerns appear:
> * a lack of diversity in the experimental setup (not enough models, not enough datasets), and
> * a lack of comparison to popular RL algorithms.
>
> We want to address these shortcomings in this general response (which we include in all individual responses), before answering each reviewer's questions separately.
>
> Our initial intent was for the article to be equally balanced between a theoretical section, a bandits section, and a section on real-world applications, which is why we chose to keep the experimental section concise. However, the reviewers' comments have helped us realize that the paper would be strengthened by the inclusion of additional experiments, and we thank them for that.
>
> We have run a series of new experiments, which we hope will fully answer their concerns. In particular, we add two additional models (Qwen 2.5 3B and Llama 3.2 3B), a much larger dataset (NuminaMath) and a comparison to GRPO. The resulting tables can be found below (sadly, a recent change in the conference's rules does not allow us to share graphs anymore):
>
> **Table 1: Training Dynamics of the off-VPG Algorithm Across Different Baselines.**
> This extends our Figure 3 with additional models and datasets (Same hyperparameters. Reported values are the maximum pass@1 accuracy (in %) on the test set observed during training (2000 steps, with evaluation every 200 steps). Some runs crash during training, as in Figure 3; these are marked in bold.)
>
> |Dataset|Model|δV=-0.5|δV=-0.3|δV=-0.2|δV=-0.1|δV=0|δV=0.1|δV=0.2|δV=0.3|
> |-------|-----|--------|--------|--------|--------|-----|-----|-----|-----|
> |MATH|Llama8B|48.7|49.1|49.3|49.3|**48.9**|**47.5**|**46.1**|**43.5**|
> ||Llama3B|45.9|46.7|46.4|46.4|**45.4**|**44.5**|**43.9**|**45.0**|
> ||Qwen3B|62.2|63.5|65.1|65.6|64.9|**37.8**|**14.7**|**13.0**|
> |NuminaMath|Llama8B|20.5|21.5|22.2|23.6|**23.1**|**17.8**|**18.0**|**17.2**|
> ||Llama3B|20.5|21.8|21.8|21.7|22.7|**20.7**|**19.8**|**19.6**|
> ||Qwen3B|20.3|37.6|40.6|42.0|40.8|**23.7**|**21.9**|**19.7**|
>
> In our article, Figure 3 provides empirical evidence of how the choice of baseline δV affects the performance of our off-VPG algorithm.
> We extend the same experiment to two additional models (Llama3B and Qwen3B) and an additional dataset (a 144k-token subset of NuminaMath)
>
> We consistently observe the same two phenomena highlighted in Figure 3:
>
> - When the baseline is large (δV ≳ 0), training becomes unstable and often crashes (bold entries in **Table 1**).
> - As the baseline increases, the training becomes faster—reflected in the average left-to-right progression of higher scores within each row (for runs that did not crash).
>
> The general appearance of the training and test accuracies are also coherent with our earlier experiments. These results reinforce the robustness of our recommendation: to use a slightly negative baseline, such as δV = -0.1.
>
> **Table 2: Comparison of off-VPG and GRPO, off-policy**
> (10000 steps, with evaluation every 200 steps)
>
> |Dataset (Train & Test)|Model|off-VPG (off-policy)|GRPO (off-policy)|
> |------------------|------|--------------------------|----------------|
> |MATH|Llama8B|52.0|50.9|
> ||Llama3B|49.0|49.1|
> ||Qwen3B|64.8|61.7|
> |NuminaMath|Llama8B|26.1|24.8|
> ||Llama3B|25.5|25.1|
> ||Qwen3B|44.3|40.8|
>
> We conduct in **Table 2** a performance comparison between GRPO and off-VPG.
> We use the GRPO variant from (DeepSeekMath: Pushing the Limits of Mathematical Reasoning in Open Language Models, Shao et al), with KL regularization coefficient β = 0.001 and symmetric clipping at 0.2.
> For off-VPG, in line with our previous conclusions, we use a fixed baseline of δV = -0.1.
>
> Under these conditions, off-VPG appears to perform slightly better on average than GRPO, despite being more simple. All training runs are stable. We also observe that off-VPG achieves faster score improvements than GRPO, especially when training Qwen3B.
>
> Note that we do not yet claim that off-VPG is universally superior to GRPO--a fully rigorous comparison is beyond the scope of this work. Rather, we argue that its performance is competitive and reliable across diverse setups (three models and two datasets), and we plan on pursuing a more systematic investigation of this comparison in future work.
>
> If the article is accepted, we plan on including two additional graphs in the main text (a variant of Figure 3 with Qwen3B, and a comparison between off-VPG and GRPO with Llama 8B), and to put the other results in the Appendix, as they do not change the article's overall narrative (though they make our conclusions more robust).
>
> Note that we are very willing to run additional experiments in the remaining discussion period, should the reviewers deem it necessary.
>
> --- End of general response ---
>
> * First, it omits some  [...] off-policy updates, and GRPO?
> * The evaluation is also [...] in the literature, such as [1].
>
> Regarding the diversity of datasets and models, as well as comparisons to other algorithms (GRPO), we refer to Tables 1 and 2 of the general response above, where it is shown that our method is competitive and that our results are robust.
> Thank you for pointing out the APA paper, which we will reference in our revised manuscript. This paper shares goals with ours. However, the approaches differ significantly in terms of methodology, theory, and experiments.
>
> * Finally, some of the theorems in the appendix seem underexplained.
>
> *Do you have particular examples in mind? We are happy to give you additional explanations, and potentially add them to the appendix.*
>
> * The original motivation [...] efficiency in any way.
>
> *In this context, efficiency should not be understood purely in terms of "final score at the end of training". Some of the practical advantages of off-policy RL, such as the capacity to handle asynchronicity between the worker Gpus (in charge of generating trajectories) and trainer Gpus (in charge of updating the weights) or the greater simplicity in implementation (no constant communication between worker Gpus and trainer Gpus), are already known to the community. Others, such as the fact that it allows several passes over the same data points (resulting in more data efficiency), have not yet been analyzed in depth in the context of LLMs, but we plan on rigorously studying these in future work.
> (For a pure score comparison, consider also Tables 1 and 2 from our general response and Table 3 in our response to reviewer oGYm: off-policy and on-policy final accuracies seem comparable, but these are preliminary experiments)*
>
> * For equation (1), how are you avoiding the importance sampling issue?
>
> *We do not face this issue, because the objective function that we optimize is NOT the expected reward $\mathrm{E}_{y \sim \pi}[r(y)]$ of the trained policy $\pi$ (though it is not far from it, as the actor policy $\mu$ is kept close to $\pi$ in the setups that we consider).
> This is what makes theorems 4.2 and 4.3 interesting, as they describe the limit policy of a gradient descent on this unusual off-policy objective function.*
>
> * How do you choose the N to [...] respect to N?
>
> *The larger the number $N$ of steps between two policy updates, the closer we are to the limit distribution $\pi_{\mu,V}^\star$ (using the notations of Thm 4.2), hence to the situation that our theory describes. In our bandits experiments, we have observed that the training curves in the policy improvement setting remain similarly shaped when we let $N$ vary, showing the robustness of our conclusions. In real world applications, the choice of $N$ is often dictated by practical concerns, such as the ratio between the number of trainer Gpus and worker Gpus available.*
>
> * Are the bounds in the bandit RL setting or the multi-step RL setting?
>
> *Theorem 4.2 applies in the case where the actor policy $\mu$ is fixed, and a gradient descent is performed on the trained policy $\pi$'s logits. Theorem 4.3 describes the policy improvement case, where the policy $\pi$ is trained on a given actor policy $\mu$, then the actor policy is updated $\mu \leftarrow \pi$ and the process is repeated. '*
>
> * Could you provide more motivation [...] is necessary.
> * How is delta V trained in practice [...].
>
> *In the bandits setting, $V$ is not trained: it is a fixed hyperparameter. The behaviour of the algorithm depends on whether $V$ is larger or smaller than $V^\mu$.
> In the LLM setting, as you rightly observe, $V^\mu = V^{\mu(\cdot |x)}$ depends on the prompt $x$. We substract a correction $\hat V + \delta V$ to each reward $r(x,y)$. $\hat V$ is an estimator of $V^{\mu(\cdot |x)}$, computed as in GRPO (for example). $\delta V$, just like $V$ in the bandit setting, is a fixed hyperparameter: it is the degree of freedom of our algorithm.
> The reward corrected by $\hat V$, i.e. $r(x,y) - \hat V$, has expected value $0$ w.r.t. to $\mu(\cdot|x)$ : $\mathrm{E}_{y \sim \mu(\cdot |x)}[r(x,y) - \hat V] = 0$ (here the notation hides the group sampling). Hence having $\delta V <0$ is the same as having $V<V^\mu$ in the bandits setting, and having $\delta V \geq  0$ is the same as having $V\geq V^\mu$ in the bandits setting (see also our answer to a similar question from Reviewer sBrs).*
>
> * For most of the bounds in the appendix, [...] to be bounded, right?
>
> We assume that the set of $y$ is finite and that $\mu$ has full support, so the ratio is necessarily bounded.
> The ratio $\pi(y) / \mu(y)$ does not appear explicitly in our derivations, but the interplay between $\pi^\star$ and $\mu$ can be seen implicitly.
> For example, if $\pi^*$ differs significantly from $\mu$, and the subtracted baseline $V$ is not conservative enough, good arms may be eliminated too early. This can be seen, e.g., in Theorem 4.2: when $V < V^\mu$, we have
> $$
>     \operatorname{supp} \pi^*_{\mu, V} = \{y \mid \mu(y)(r(y) - V) - \tau_{\mu, V} > 0\}.
> $$

---

> > ### Comment · Reviewer_Wxir · 2025-08-08
> >
> > I thank the authors for their thorough response. I will keep my score.

---

> > > ### Author Response · Authors · 2025-08-08
> > >
> > > Dear reviewer,
> > >
> > > In response to the weaknesses that you had listed, we have included new models and tasks to our experiments, as well as a comparison to a baseline. Is there any other concern we haven’t yet addressed that, if answered, would make you consider raising your score?

---

### Note · Authors · 2025-08-12

We would like to thank all reviewers for their thoughtful feedback.
We appreciate the positive reception of our work and are grateful for the valuable recommendations that have helped us strengthen our contribution.
The main concern expressed in the reviews was the relative lack of diversity and comparisons in our experimental setup: it helped us realize that this was indeed a weak point of our work.
Consequently, we conducted and presented a series of new experiments featuring new models and new datasets. The outcomes of these experiments are consistent with our earlier findings and reinforce the robustness of our results.
We think that these additions will greatly improve the credibility of the paper (as the new results align with our previous conclusions, integrating them into the manuscript will also be straightforward).
We are encouraged that the reviewers responded positively to these new experiments and expressed satisfaction with the revisions.
While those points are slightly outside the primary scope of our work, we also conducted some comparisons to GRPO and some preliminary experiments in the on-policy setting, which will enrich the experimental section and pave the way for future research. We thank the reviewers once again for these insightful suggestions, as well as for their other helpful comments.

Best regards,

The authors

---

### Decision · Program_Chairs · 2025-09-17

**Decision:**

Accept (poster)

**Comment:**

The paper analyzes a simple off-policy policy gradient algorithm, showing both theoretically and empirically that emphasizing positive rewards while down-weighting failures yields policy improvement guarantees and more stable performance when fine-tuning LLMs and in bandit settings. The reviewers generally recognized the novelty of the theoretical analysis and appreciated the empirical validation, though some raised concerns about the limited diversity of experiments and missing comparisons to key baselines The additional experiments and comparisons added during rebuttal helped address some of these concerns, but the paper would benefit from a more candid discussion of its limitations. In particular, the reliance on bandit-style analyses and the narrow experimental scope leave open questions about generalization. The authors are encouraged to explicitly acknowledge these limitations and discuss how future work could broaden the empirical validation and strengthen connections to on-policy baselines.